# Contribution of epigenetic variation to adaptation in *Arabidopsis*

Marc W. Schmid [1,2,3,7], Christian Heichinger[1,2,8], Diana Coman Schmid[1,2,9], Daniela Guthörl[1,2], Valeria Gagliardini[1,2], Rémy Bruggmann[4], Sirisha Aluri[5], Catharine Aquino[5], Bernhard Schmid [2,6], Lindsay A. Turnbull [2,6,10] & Ueli Grossniklaus [1,2]

In plants, transgenerational inheritance of some epialleles has been demonstrated but it remains controversial whether epigenetic variation is subject to selection and contributes to adaptation. Simulating selection in a rapidly changing environment, we compare phenotypic traits and epigenetic variation between *Arabidopsis thaliana* populations grown for five generations under selection and their genetically nearly identical ancestors. Selected populations of two distinct genotypes show significant differences in flowering time and plant architecture, which are maintained for at least 2–3 generations in the absence of selection. While we cannot detect consistent genetic changes, we observe a reduction of epigenetic diversity and changes in the methylation state of about 50,000 cytosines, some of which are associated with phenotypic changes. Thus, we propose that epigenetic variation is subject to selection and can contribute to rapid adaptive responses, although the extent to which epigenetics plays a role in adaptation is still unclear.

[1] Department of Plant and Microbial Biology, University of Zurich, Zollikerstrasse 107, 8008 Zurich, Switzerland. [2] Zurich-Basel Plant Science Center, University of Zurich, ETH Zurich and University of Basel, Tannenstrasse 1, 8092 Zurich, Switzerland. [3] Service and Support for Science IT, University of Zurich, Stampfenbachstrasse 73, 8006 Zurich, Switzerland. [4] Interfaculty Bioinformatics Unit and Swiss Institute of Bioinformatics, University of Bern, Hochschulstrasse 6, 3012 Bern, Switzerland. [5] Functional Genomics Center Zurich, ETH and University of Zurich, Winterthurerstrasse 190, 8057 Zurich, Switzerland. [6] Department of Evolutionary Biology and Environmental Studies, University of Zurich, Winterthurerstrasse 190, 8057 Zurich, Switzerland. [7]Present address: MWSchmid GmbH, Möhrlistrasse 25, 8006 Zurich, Switzerland. [8]Present address: L. Hoffmann-La Roche AG, Grenzacherstrasse 124, 4070 Basel, Switzerland. [9]Present address: Scientific IT Services, ETH Zurich, Weinbergstrasse 11, 8092 Zurich, Switzerland. [10]Present address: Department of Plant Sciences, University of Oxford, South Parks Road, Oxford OX1 3RB, UK. These authors contributed equally: Marc W. Schmid, Christian Heichinger. Correspondence and requests for materials should be addressed to U.G. (email: grossnik@botinst.uzh.ch)

Modifications of DNA and chromatin are epigenetic marks that affect gene expression and play an important role in plant development and responses to the environment. In contrast to mammals, the germline of plants is not set aside early during development but forms only later when somatic cells are committed to form gametes[1]. Thus, epigenetic marks, for example DNA methylation that changes during development or is affected by environmental conditions, are potentially heritable. Consequently, such marks may also play a role in adaptive responses to a changing environment. Furthermore, changes in DNA methylation occur much more frequently than genetic mutations[2,3]. It thus seems plausible that, modulated by genomic context, epimutation rates may be high enough to rapidly uncouple epigenetic from genetic variation, yet low enough to sustain a response to selection[4].

Important for a potential role of epigenetics in adaptation are the extent of environmental effects on epigenetic variation, the rate of spontaneous epimutations, and the effect of epigenetic variation on the phenotype. Furthermore, epigenetic variation and the associated phenotype must be heritable and uncoupled from genetic variation. While genome-wide patterns of anonymous DNA methylation markers were found to be associated with distinct environments[5,6], genome-wide studies with single base-pair resolution in *Arabidopsis* have revealed extensive variation in DNA methylation patterns between different populations and accessions[7,8]. This variation was partly due to different environments; however, it was mostly linked to underlying genetic differences in *cis* and also affected by major *trans*-acting loci[7,8]. Nonetheless, recent work on epigenetic recombinant inbred lines (epiRILs) in *Arabidopsis* suggests a significant contribution of genetically induced epialleles to phenotypic variation, which is independent of genetic variation[9,10]. However, the role of epigenetic variation in adaptation is currently not well understood and its evolutionary relevance remains highly controversial[11–16].

To investigate whether epigenetic variation has the potential to confer a selectable fitness advantage, we used material from a previously conducted selection experiment that simulated a fragmented habitat subject to frequent disturbance[17]. In this experiment, *Arabidopsis* populations were grown in discrete patches and only seeds that dispersed to new locations contributed to the next generation, simulating a "dynamic landscape". The experiment started with a population consisting of 19 distinct *Arabidopsis* genotypes (recombinant inbred lines (RILs) derived from the accessions Cape Verde Islands (Cvi) and Landsberg *erecta* (L*er*)[18]. After five generations, genetic variation was strongly reduced and only two genotypes (CVL39 and CVL125) dominated the populations[17]. In the three populations assessed at that time (landscapes D1/D5/D6), these two genotypes represented 93% (D1), 97% (D5), and 79% (D6) of the populations. On average, the two genotypes were similarly present (CVL39: 47%; CVL125: 43%), but their proportions among the three replicated, and thus independently selected, populations varied (CVL39: 33%, 87%, and 21% and CVL125: 60%, 10%; 59% in populations D1, D5, and D6, respectively).

To examine whether there had also been selection of epigenetic variation within the two dominant genotypes, we compared phenotypic traits and genome-wide DNA methylation states of progeny from the original founder population ("ancestral", D0) with three independently selected, replicate populations after five generations of selection ("selected", D1/D5/D6). To do this, individuals from the ancestral populations (D0) and the three selected populations (D1/D5/D6) from each of the two genotypes (CVL39/CVL125) were grown together for three generations (A1/S1, A2/S2, A3/S3) in a randomized matrix and controlled environment. To eliminate confounding maternal effects,

phenotypes were only measured in the second and third generation (Fig. 1a and Supplementary Fig. 1 for more details). To assess whether genetic differences between ancestral and selected individuals might have contributed to phenotypic differences, we also resequenced nine individuals from one genotype (at least two from each population, Supplementary Fig. 1).

We find that traits related to reproduction are significantly different even in the third generation after selection, indicating inheritance of the acquired phenotypes. While we cannot detect any significant genetic changes, we observe a reduction of epigenetic diversity and changes in the DNA methylation patterns between ancestral and selected populations. Differences in DNA methylation and gene expression between ancestral and selected populations are found in pathways relevant to the altered phenotypic traits, e.g., flowering time. We also identify a potential epiallele of a non-coding RNA, which might contribute to the observed phenotypic differences in one of the two genotypes. DNA methylation and expression of this gene are negatively correlated in all plant individuals. Low DNA methylation and high expression are further associated with delayed flowering. These correlations are also apparent in natural *Arabidopsis* accessions. Finally, we discuss the origin of the selected epigenetic variation and conclude that hybridization between the two natural accessions, which is the basis to generate RILs, also contributes to the epigenetic variation observed in our study.

## Results

**Differentiation of phenotypic traits during selection**. We found that traits related to reproduction and fecundity, i.e. the day of bolting, the number of rosette leaves at bolting, the number of branches, and the number of siliques, differed significantly between ancestral and selected populations in both genotypes (all $P < 0.001$, ANOVA based on 264 individuals in total, Fig. 1b, Supplementary Data 1, Supplementary Fig. 1). Specifically, bolting was delayed and the number of rosette leaves was increased in selected compared to ancestral populations, such that individuals from selected populations flowered on average later than the ones from ancestral populations (Fig. 1b). We also observed a significant increase in the number of branches and the number of siliques in selected compared to ancestral populations. Increased branching has been shown to increase the average seed dispersal distances in *Arabidopsis*[19]. Likewise, an increased number of siliques, which is correlated with overall seed production[18], will augment the chances of seeds being dispersed to distant sites. This is consistent with the observation that efficient seed dispersal has been the primary selective force in dynamic landscapes[17]. Assuming phenotypic variation within each RIL is explained by epigenetic differences, and the variation between RILs is explained by genetic differences (see below), epigenetic differences explained almost half as much of the variance of traits related to reproduction and fecundity compared to genetic differences. Whereas the contrast comparing ancestral with selected individuals explained on average around 7.1% of the total sum of squares, the contrast comparing the two RILS to each other explained on average 15.4% of the total sum of squares (Supplementary Data 1). It is likely that this rather large contribution of epigenetic relative to genetic variation reflects the fact that the genetically caused phenotypic variation of populations growing in the dynamic landscapes was strongly reduced due to selection[17], i.e., that the two dominant RILs were phenotypically already very similar to each other.

**Reduction of epigenetic diversity during selection**. To investigate whether these heritable phenotypic changes were paralleled by changes at the level of DNA methylation, we examined

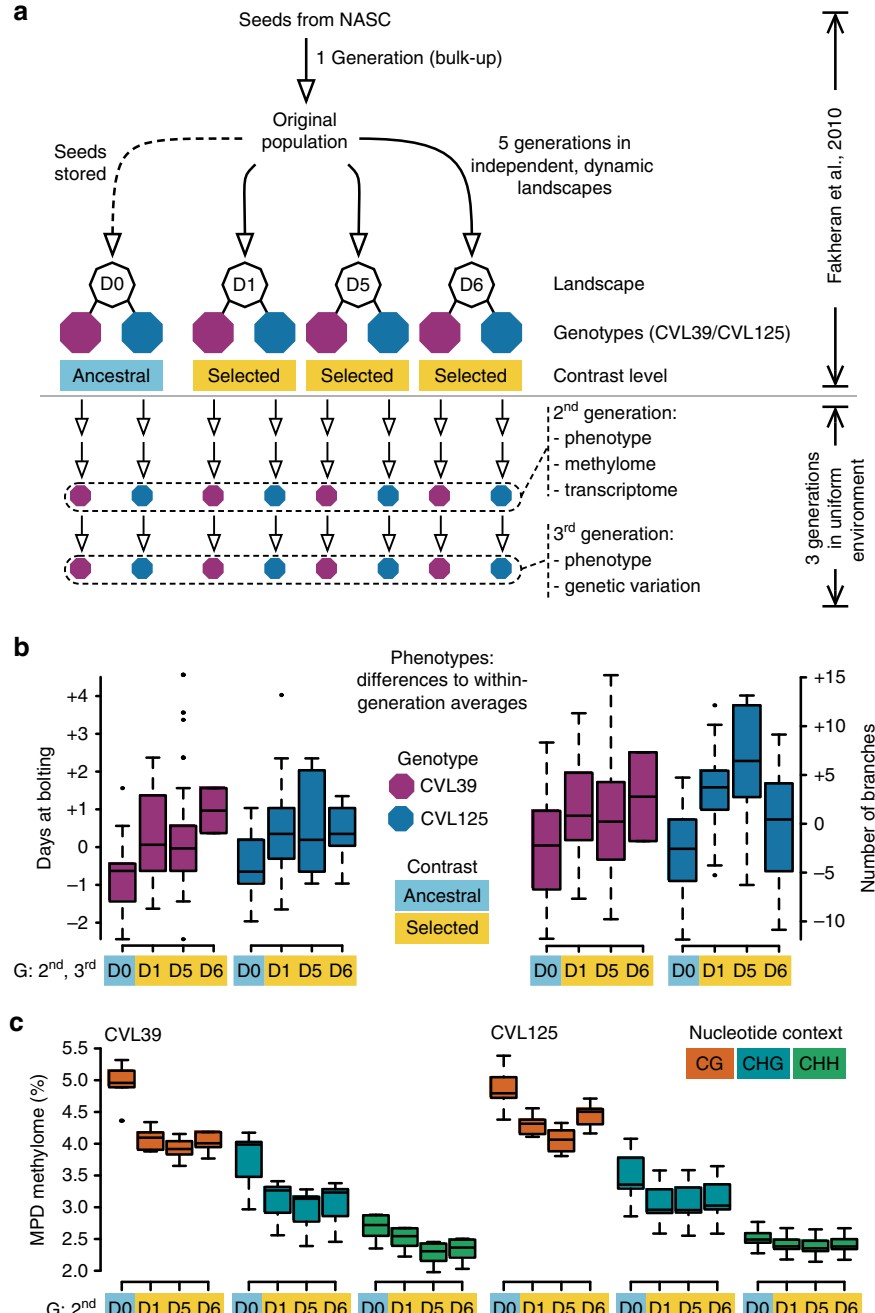

**Fig. 1** Experimental design used to demonstrate adaptive traits and reduced epigenetic diversity after selection. **a** Schematic representation of the experimental design (see also Supplementary Fig. 1 for more details). The original population consisted of 19 equally represented genotypes grown for five generations in a selective environment[17]. Two genotypes (RILs CVL39 and CVL125) dominated the selected populations and were used in the present study. Offspring from the original population (D0) and selected populations (originating from three independent experiments, i.e., landscapes D1, D5, and D6) were grown for three generations in a non-selective environment (controlled conditions and randomized plant locations). Phenotypes were measured in the second and third generation. Methylome and transcriptome were profiled in the second generation. Genomes were resequenced in the third generation. **b** Comparison of phenotypic traits of offspring of the ancestral (D0) and selected populations (D1/D5/D6). In the second and third generation, flowering time was significantly delayed and the number of secondary inflorescences was significantly increased in the selected compared to the ancestral populations (Supplementary Data 1). To show both generations at once, the numbers shown are differences to the averages across all populations of a genotype within a given generation. **c** The mean pairwise distance (MPD) in DNA methylation patterns between the individuals of a given population reflects the epigenetic diversity within the population. In the second generation, epigenetic diversity was consistently higher in the ancestral populations (D0) compared to the selected populations (D1/D5/D6)

genome-wide DNA methylation levels of ancestral and selected populations in the second generation in a common environment (A2/S2). For each selected population (D1/D5/D6) from each of the two genotypes (CVL39/CVL125), four plants were sequenced

(3*2*4 = 24 in total) and compared to 16 individual plants from the ancestral populations (8 per genotype, Supplementary Fig. 1, Supplementary Data 2). To characterize the epigenetic diversity within the different populations, we calculated the mean pairwise

distance (MPD) within the ancestral and the selected populations as a measure that is not correlated to sample size[20]. Epigenetic diversity measured by MPD was consistently lower in selected compared to ancestral populations (Fig. 1c), which could be explained by the selection of favorable epigenetic variants. We then compared the methylation levels at over 80% of all cytosines in the genome and identified 49,084 cytosines with consistent and significant differences between ancestral and selected populations (Supplementary Data 3, 4). Most of these differentially methylated cytosines (DMCs) were specific to each genotype (30,567/16,863 in CVL39/CVL125, and 827 shared by both).

**Functional characterization of genes with DMCs.** DMCs were distributed across the entire genome (Supplementary Fig. 2). To test whether DMCs formed specific clusters, we compared the distances between DMCs to distances between randomly sampled Cs. The DMC distribution exhibited an enrichment of short distances (below 100 bp) between neighboring DMCs compared to randomly sampled Cs (Fig. 2a). Consequently, defining differentially methylated regions (DMRs; as in refs[2,21]), resulted in 522/230 regions with 3750/1476 DMCs (12/8%) and an average size of 67/65 bp in CVL39 and CVL125, respectively (Supplementary Data 5, 6). However, such data-driven DMR definitions are highly parameter-dependent (Fig. 2b). Therefore, we used genomic loci (e.g., genes) to summarize DMC occurrences and changes in DNA methylation levels (Fig. 2c, Supplementary Data 7, 8). Loci were frequently associated with several DMCs. For instance, 254 loci had at least 10 DMCs and an average methylation change of 50% (156/98 in CVL39/CVL125). To functionally characterize the genes associated with DMCs, we tested for enrichment of gene ontology (GO) terms over a wide range of different thresholds for the number of DMCs per gene and the average change in methylation levels. We found 20 and 23 GO terms that were robustly enriched in CVL39 and CVL125, respectively (i.e., significant in at least 50 out of 121 parameter combinations, Supplementary Data 9). For example, genes involved in the change of vegetative to reproductive growth (GO:0010228), were enriched in both genotypes (Supplementary Data 10). Interestingly, the genes under this GO term, for instance FPA with 77 CG-DMCs in CVL125, are also involved in the regulation of flowering time, which is one of the traits that were different between ancestral and selected populations.

**Differences in DNA methylation vary by sequence context.** In *Arabidopsis*, cytosine methylation can be found in three different sequence contexts: CG, CHG, and CHH, where H denotes A, C, or T. DMCs were highly enriched in the CG context (90/93% in CVL39/CVL125) compared to the genome-wide distribution of cytosines in *Arabidopsis* (13%, Fig. 2d). This enrichment was consistent with previous reports, where changes in DNA methylation over multiple generations were mostly limited to the CG context[2–4,21]. With recent estimates for forward (methylation) and backward (demethylation) epimutation rates, a loss of DNA methylation in the CG context would be expected[4] (except for CGs in transposons). However, methylation at CG-DMCs was on average higher in the selected populations of CVL39 and only marginally lower in the selected populations of CVL125 (Fig. 2e). Given that most CG-DMCs were located within genes (preferentially towards the 3′ end, Fig. 2f, g), this observation may suggest that epimutation rates vary widely between different genotypes and evolutionary histories.

Differences in DNA methylation levels at DMCs varied depending on the context (Fig. 2e). Average differences in CVL125 were 68.2%, 35.8%, and 23.0% in the CG, CHG, and CHH contexts, respectively. Average differences were similar in

CVL39 with 65.5%, 37.0%, and 22.1% in the CG, CHG, and CHH contexts, respectively (Fig. 2e, Supplementary Data 3, 4). However, these differences suggest that most of the DMCs were not fully methylated in one and fully demethylated in the other condition. Cytosines with such a binary methylation state could be interpreted as epigenetic variation that resembles genetic variation, for example a single-nucleotide polymorphism (SNP), which has at maximum four different states (A, T, C, G). Thus, to estimate how many cytosines exhibit DNA methylation patterns that resemble a SNP (i.e., which are either completely methylated or demethylated), we counted the number of cytosines with DNA methylation levels below 5% or above 95% in all individuals of each RIL. Only 0.0013% and 0.0017% of all cytosines in CVL39 and CVL125, respectively, fulfilled this criterium (almost exclusively in the CG context). Thus, almost none of the cytosines displayed a DNA methylation pattern that resembles a SNP. It is likely that this is because the methylome data had been derived from inflorescences, which consist of many different tissues and cell types. As reported recently, tissue and cell type-specific methylomes can be very distinct[22–25]. However, for DNA methylation variants to be inherited, only the cells contributing to the reproductive lineage are required to maintain that variant. Interestingly, several DNA methyltransferases involved in DNA methylation are expressed at high levels in the stem cell niche of the shoot apical meristem (data from Yadav et al.[26]). *METHYL-TRANSFERASE1 (MET1, At5g49160)*, required for maintenance of CG methylation, was expressed at the 97th percentile. Similarly, *CHROMOMETHYLASE3 (CMT3, At1g69770)*, maintaining DNA methylation in the CHG context, was expressed at the 90st percentile, while *CHROMOMETHYLASE2 (CMT2, At4g19020)*, involved in the maintenance of CHH methylation, could not be assessed because it is not represented on the ATH1 microarray used by Yadav et al[26]. However, *DOMAINS REARRANGED METHYLTRANSFERASE2 (DRM2, At5g14620)*, which is involved in the RNA-dependent DNA methylation (RdDM) pathway, which also controls methylation in the CHH context, was expressed at high levels (78th percentile). High expression of these DNA methyltransferases may result in stable DNA methylation levels in the stem cells of the shoot apical meristem. Thus, it would be interesting to determine whether stem cells have reduced epimutation rates because this could explain faithful inheritance of epigenetic variation even if it was variable between different tissues and cell types.

**CHG/CHH-DMCs are frequently located in RdDM target regions.** To determine whether DMCs affected genes or other specific regions of the genome, we mapped them to genomic features. DMCs in the CHG and CHH contexts were mostly limited to transposons, and methylation levels were on average higher in selected compared to ancestral populations (Fig. 2e, f). An exception were the CHG-DMCs with reduced average methylation levels in the selected populations of CVL125, which also occurred frequently in genic regions (Fig. 2f). Around one-third of these CHG-DMCs (111/356) were located within a 3.3 kb region on chromosome 2 within the gene At2g25050 (encoding the actin-binding formin homology FH2 protein). However, the functional relevance of these DNA methylation changes remains unclear as expression of the gene seems unaffected (Supplementary Data 11), at least at the time-point the transcriptome was measured. Nonetheless, DNA methylation of transposons may be involved in the regulation of neighboring genes[3,27]. In *Arabidopsis*, transposons are silenced by 24-nucleotide-long siRNAs (24-nt siRNAs) through the RdDM pathway[27]. RdDM ensures sequence-specific, stable methylation at its target regions, and it has been shown that spontaneous epimutations are much less frequent in transposons and 24-nt siRNA

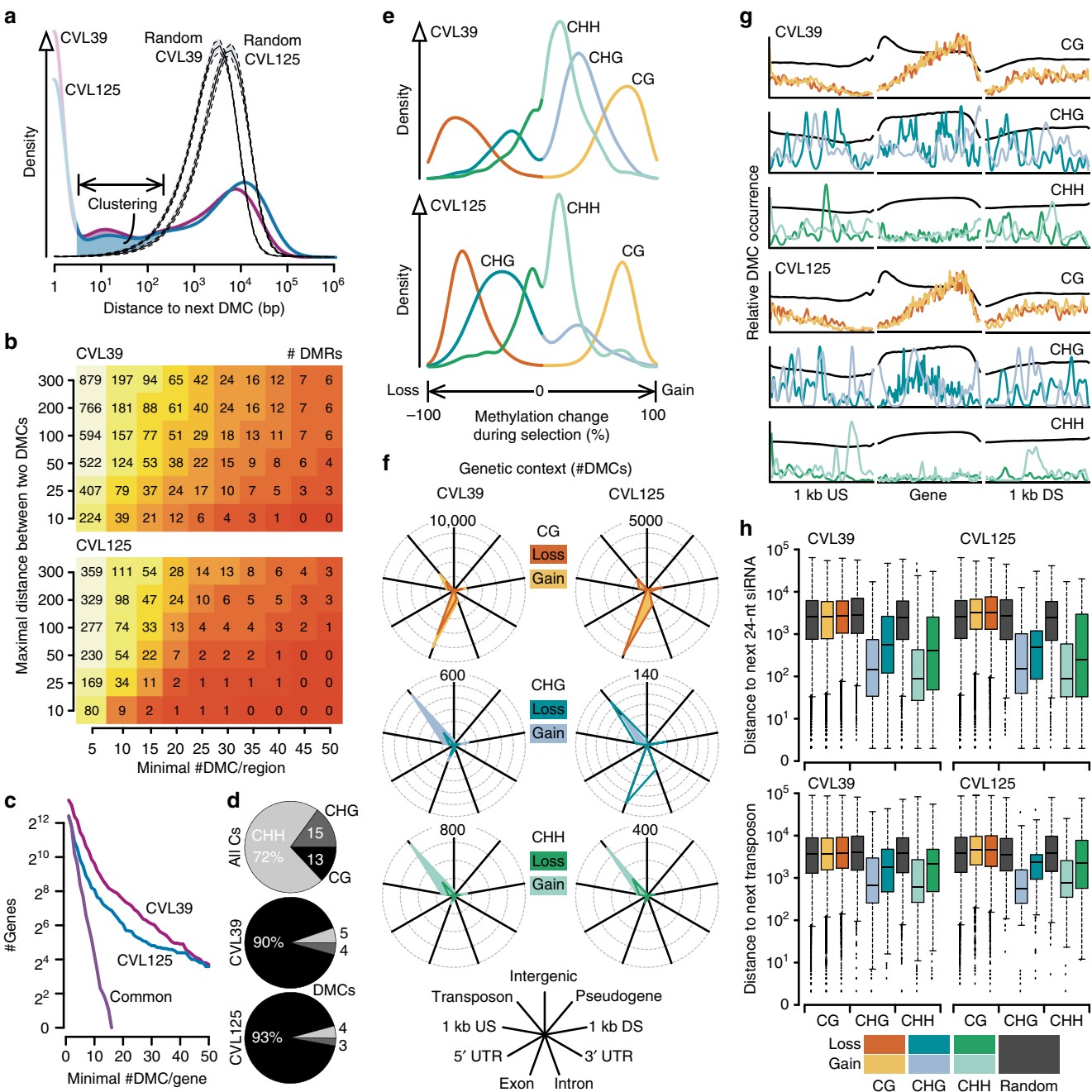

**Fig. 2** Distribution of DMCs across the genome and genomic features. **a** DMCs were found in clusters along the genome. Distances between two neighboring DMCs within such clusters were significantly smaller than expected by chance (random sampling). Distances of 1 and 2 are shaded as a large fraction of them represents DMCs in symmetric contexts (CG and CHG). **b** The number of DMRs identified by a certain minimal number of DMCs per region and a maximal distance between two neighboring DMCs for consolidation into one DMR. **c** Abundance of DMCs within genetic loci (genes and 1 kb flanking regions). **d** Proportions of DMCs within the CG, CHG, and CHH contexts. Whereas cytosine residues (all Cs) are observed predominantly in the CHH context, DMCs are preferentially located in the CG context. **e** Differences in methylation levels at DMCs. Note that methylation levels refer to population averages. The changes are therefore not synonymous to epimutations. Instead, they more likely reflect the selection of certain epigenotypes that were already present in the ancestral populations. **f** The number of DMCs found in a certain genomic context. CG-DMCs are preferentially located in gene bodies, whereas CHG/CHH-DMCs are mostly limited to transposons (see text for the genic CHG-DMCs of CVL125). US/DS: upstream/downstream flanking regions, UTR: untranslated regions. **g** The average distribution of DMCs along protein-coding genes and their flanking regions (1 kb). Black lines are all Cs in the genome. **h** The distance of DMCs to 24-nt siRNA target regions and transposons. Irrespective of the differences in methylation levels between selected and ancestral populations, CHG/CHH-DMCs were significantly closer to the potential RdDM target regions than expected based on random sampling

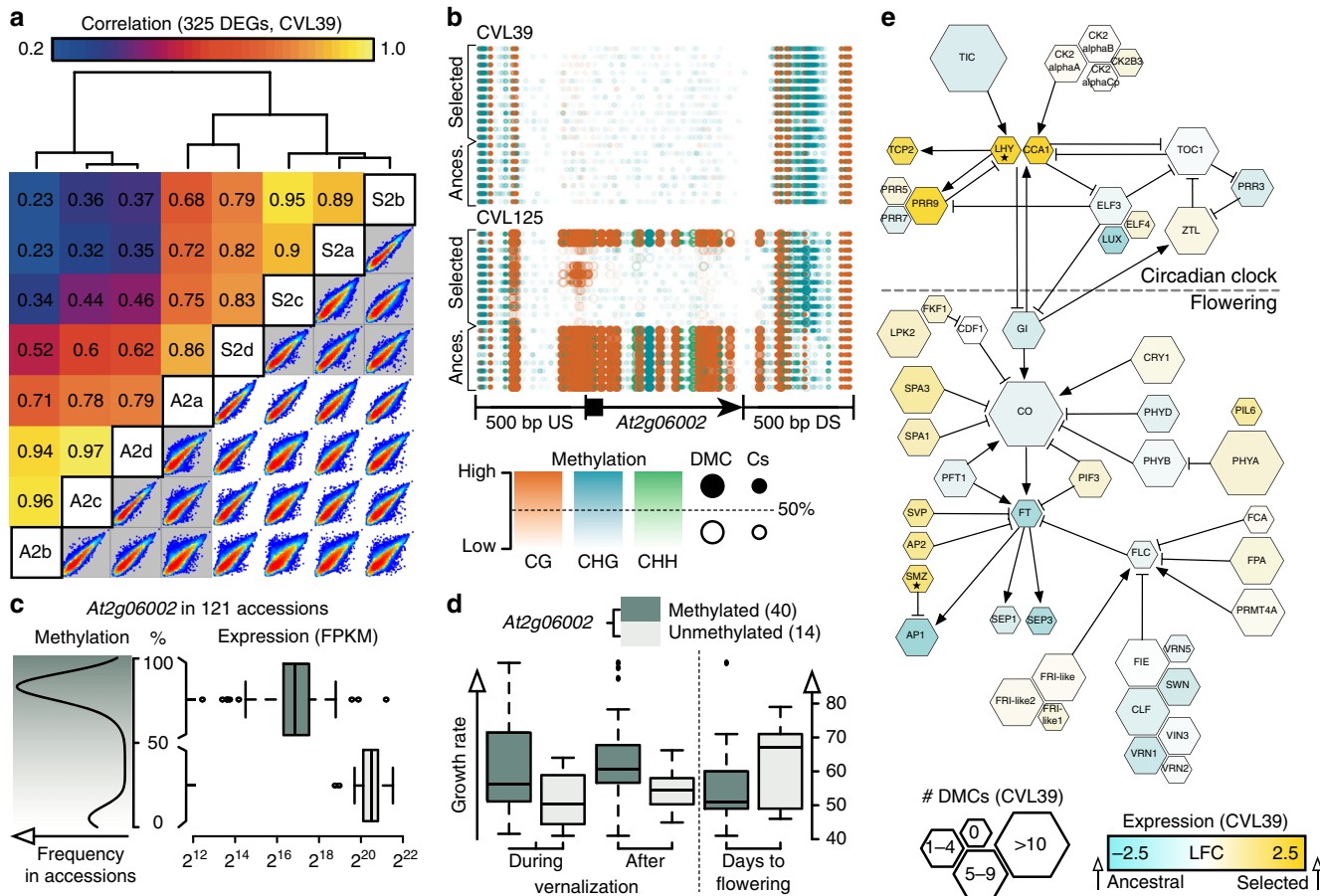

**Fig. 3** Differential gene expression links differences in DNA methylation to adaptive traits. **a** Sample correlation based on 325 differentially expressed genes (DEGs) in CVL39 and pairwise comparison of gene expression values (all genes). Samples were clustered using Pearson correlation and hierarchical agglomerative clustering (complete linkage). The individuals from the ancestral (A2) and the selected (S2) population were not perfectly separated into two groups. Intermediate individuals may reflect remaining variation within the population and indicate the selection of epigenetic variation that was already present in the ancestral population. **b** The DNA methylation profiles of *At2g06002* and its upstream/downstream (US/DS) flanking regions. Methylation levels were strongly reduced in the selected populations of CVL125 (59 DMCs with an average reduction of 47%) and expression of *At2g06002* was 37-fold higher than in the ancestral population of CVL125. **c** Methylation and expression of *At2g06002* in 121 different accessions of *Arabidopsis* was strongly correlated. Accessions were separated into two groups with either high methylation and low expression levels or vice versa. The 13-fold difference in gene expression was highly significant ($P < 10^{-15}$, two-sided *t*-test adjusted for multiple testing). **d** Growth rates during and after vernalization and days to flowering (at 10 °C) were significantly ($P < 0.05$, two-sided *t*-test) different between accessions with high/low methylation of the gene *At2g06002*. Analyses shown in **c, d** are based on data from Schmitz et al.[31] and Atwell et al.[32], see also Supplementary Data 14. **e** Genes in the circadian clock and downstream flowering time pathway were rarely identified as differentially expressed (marked with asterisk). However, both core circadian clock genes (*LHY\** and *CCA1*) were more than four-fold upregulated in the selected populations of CVL39. In parallel, major flowering-promoting genes (*GI*, *CO*, and *AP1*) and, most prominently, *FT* encoding the florigen, showed reduced expression levels. See also Supplementary Data 13

target regions compared to genes[2,4]. However, CHG/CHH-DMCs co-localized more frequently with transposons and 24-nt siRNA target regions ($P < 0.05$) and were otherwise on average closer to these regions ($P < 0.002$) than expected by chance (500 times random sampling, Fig. 2h). This finding suggests that RdDM may reinforce DNA methylation at positions with initially small differences between individuals.

**Differences in DNA methylation and gene expression.** Differential DNA methylation can affect gene expression[2,28,29]. To identify genes that potentially contribute to the observed phenotypic differences, we compared the transcriptomes of the ancestral and selected population D1 for each of the genotypes. Differences in gene expression between ancestral and selected populations were moderate and only few genes were significantly differentially expressed (325 and 1 genes in CVL39 and CVL125,

respectively). The changes in CVL39 were nonetheless surprising, given that a recent study reported that <3% of all genes are differentially expressed upon mutation of genes important for various DNA methylation pathways[28]. In comparison, the 325 differentially expressed genes correspond to around 1% of all genes. Out of the 325 candidate genes in CVL39, only 94 (27%) were associated with DMCs. Similar to previous reports[30], we did not find a global correlation between DNA methylation and gene expression. A possible reason may be that the transcriptome we determined provides a single snapshot in development and that correlations occurring at a different developmental stage or within a specific tissue were missed. However, DNA methylation and expression were linked in some specific cases. An example is the differentially expressed gene *At2g06002* (a non-coding RNA, Fig. 3b) in CVL125. This gene exhibited extreme differences in DNA methylation (59 DMCs, an average decrease of 47% in selected populations) and expression (37-fold increase in the

selected population). In CVL39, methylation levels were low in all individuals and its expression was 81 times higher than in the ancestral CVL125 population, suggesting that low methylation correlated with high expression. To substantiate this finding, we additionally analyzed previously published methylation and transcriptome data from 121 different *Arabidopsis* accessions[31] and checked whether this was a general pattern. The accessions clearly separated into two groups with either very low or high average methylation levels at the DMC positions within *At2g06002*, and expression of the gene was in average indeed 13-fold higher in the accessions with low methylation levels ($P < 10^{-15}$, two-sided moderated $t$-test adjusted for multiple testing, Fig. 3c). We further tested whether there were any phenotypic differences between the accessions from these two groups (phenotypic data was available for 40/14 accessions with low/high methylation from Atwell et al.[32]). Accessions with low methylation (high expression) had a reduced growth rate (during and after vernalization) and flowered on average 7 days later than the accessions with high methylation (low expression) levels (at 10 °C, no significant differences in flowering time were found at 16 °C or at 22 °C, Fig. 3d). Interestingly, this resembles the case of CVL125, where methylation levels were lower, expression levels were higher, and flowering was delayed in selected populations compared to the ancestral population.

***At2g06002*, a novel epiallele involved in flowering?** To verify that the expression of *At2g06002* correlated with its methylation status, we monitored its expression with droplet digital PCR (ddPCR) in the same CVL125 individuals that were used to generate the DNA methylation data (four individuals per population). Indeed, expression of *At2g06002* was significantly higher in the selected populations (fold-change = 8.5, $P = 0.0056$, two-sided $t$-test adjusted for multiple testing). This was true for all individuals of the selected populations, except for the two individuals with a highly methylated allele. High methylation in these two individuals was associated with low expression, as observed in the individuals from the ancestral population (Fig. 3 and Supplementary Fig. 3 ). Thus, DNA methylation and expression of *At2g06002* were clearly correlated both at the individual and the accession level.

A possible mechanism linking *At2g06002* to flowering time may involve its localization in the promoter of the neighboring gene, which encodes the FRIGIDA INTERACTING PROTEIN1 (FIP1)[33]. The interactor of this protein, FRIGIDA, is a major determinant of natural variation in flowering time in *Arabidopsis*. Thus, in addition to *At2g06002*, we monitored expression of the neighboring genes (*At2g06000* and *FIP1*), *FRIGIDA*, and the florigen-encoding gene *FT* (Supplementary Fig. 3 and Supplementary Data 12). We could not observe significant differences in expression of *At2g06000*, which is the gene upstream of *At2g06002*. However, expression of *FIP1*, the gene downstream, was significantly higher in individuals of the selected populations (fold-change = 1.8, $P = 0.0019$, two-sided $t$-test adjusted for multiple testing). Expression of *FRIGIDA* was not significantly different between ancestral and selected populations ($P = 0.5147$, two-sided $t$-test adjusted for multiple testing). However, this does not exclude the possibility that *At2g06002* has an impact on flowering time through FIP1 and FRIGIDA because the interaction between FRIGIDA and FIP1 occurs at the protein level[33]. Accordingly, expression of the florigen-encoding gene *FT* was still significantly reduced in the selected populations (fold-change = 0.5, $P = 0.0244$, two-sided $t$-test adjusted for multiple testing), supporting the observation that selected populations flowered later than the ancestral population. Nonetheless, whether and how *At2g06002* is mechanistically involved in the regulation of flowering time remains to be elucidated.

**Differential regulation of the flowering time pathway.** Given the differences in flowering time between ancestral and selected populations, we specifically focused on the flowering time pathway in CVL39. Only few genes involved in this pathway showed significant differences in expression but many were associated with DMCs (Fig. 3e, Supplementary Data 13). Both of the core circadian clock genes *LHY* and *CCA1* were more than 4-fold upregulated in the selected populations of CVL39. Increased expression of these genes had previously been shown to delay flowering time[34,35]. In parallel, major flowering-promoting genes (e.g., *GI*, *CO*, and *AP1*), including the florigen-encoding gene *FT*, exhibited reduced expression levels in the selected populations. The delayed flowering time of selected populations may therefore be explained by moderate changes in the expression of regulators in the flowering time pathway, eventually leading to a reduced level of *FT*, which itself is not associated with DMCs.

**Resequencing shows extremely low genetic variation.** To investigate whether genetic changes in selected populations might have contributed to the observed changes in phenotypic traits and DNA methylation, we sequenced the genomes of two ancestral and seven selected individuals from CVL39 (including at least two individuals from each of the independently selected replicate populations). We could not find any novel transposon insertions, but we identified 14 SNPs (Supplementary Table 1), out of which 12 exhibited residual heterozygosity in the ancestral populations and were thus likely segregating during the experiment. The alleles of the two remaining SNPs, which occurred in individuals of all three selected populations, could not be found among the 20 individuals of the ancestral population we tested. However, it is extremely unlikely that these two SNPs represent novel mutations that arose during the selection experiment in all three independently selected replicate populations, given the low rates of spontaneous genetic mutations in *Arabidopsis*[36]. Although we cannot exclude that these segregating SNPs contributed to the observed phenotype or differences in DNA methylation patterns, it seems highly unlikely that they were the sole cause because none of the affected genes (or genes near an intergenic SNP) have a known role in processes related to flowering time, plant stature, or DNA methylation.

**Discussion**

Our study suggests that epigenetic variation within populations of *Arabidopsis* can be subject to selection and contribute to adaptation. Offspring of ancestral and selected populations grown together in a controlled environment exhibited significant phenotypic differences even in the second and third generation after the selection experiment was completed. The observed phenotypic differences were paralleled by an overall reduction of epigenetic diversity in the selected populations and by significant changes in DNA methylation levels at individual cytosines throughout the genome. In contrast to the expectation based on previous studies on de novo acquired epimutations, methylation of DMCs in the CG context was on average higher after five generations of selection and CHG/CHH-DMCs were significantly enriched in regions targeted by the RdDM pathway. We observed an overall reduction in epigenetic diversity, which indicates that certain epigenetic variants were selected during the course of the experiment. However, it is difficult to discern the origin of the selected epigenetic variation, as it could have been present at low frequency in the population prior to the experiment (standing epigenetic variation) or acquired during the selection experiment. Two frequently discussed sources of epigenetic variation are random, spontaneous de novo epimutations and environmentally induced epimutations[16]. Although random epimutations occur in

every generation, the selected ones most likely arose before the selection started because random epimutations would have had to arise several times independently in replicated selection experiments. Even though spontaneous epimutation rates are higher than genetic mutation rates, this seems unlikely. Although environmental conditions have been shown to induce changes in the methylome (e.g., refs [21,37,38]) and epigenetic alterations caused by stress treatments can prepare the plant for future stress periods (reviewed by Bäuerle[39]), it remains largely unknown whether and to which extent environmentally induced epigenetic variation can be inherited through sexual reproduction[14,40–42]. For example, recent studies provide direct or indirect evidence for reprogramming of the epigenome during gametogenesis[23,25,43–46] or embryogenesis[24]. Thus, inheritance of environmentally induced epigenetic variation may be limited and restricted to certain regions of the genome.

In our particular study, it is likely that hybridization, which is the basis to generate RILs, contributed to the epigenetic variation in the ancestral population. Hybridization results in higher epimutation rates at certain genomic regions, with a bias towards the state of one parent, and such changes, e.g. in DNA methylation, can be heritable[47,48]. In general, we observed that DNA methylation patterns in the RIL resembled the DNA methylation pattern of the original accessions in *cis*, i.e., that genomic regions inherited from one accession had a DNA methylation pattern that was overall more similar to the contributing accession than the other (Supplementary Fig. 4). An exception was CHH methylation in pericentromeric regions, which in both RILs was more similar to Cvi than to Ler, potentially indicating a *trans*-effect from one or more Cvi alleles. Similarly, individual loci may be in an unstable methylation state that is caused by opposing *cis*- and *trans*-effects. An example for such a case may be the gene *At2g06002*. In CVL39, the gene originates from Cvi but in CVL125 it stems from Ler. In the parental accessions, the Cvi allele is demethylated as in all individuals of CVL39 and the Ler allele is strongly methylated as in the ancestral individuals and two selected individuals of CVL125 (Supplementary Fig. 2 for the genotype, methylation data from neomorph.salk.edu/1001_epigenomes.html). Unless the Ler allele was actively demethylated during the hybridization event, the methylated allele was likely the original state in CVL125. If this were the case, it may be possible that the methylated Ler allele in CVL125 lost its methylation because a *trans*-acting factor necessary for the maintenance of DNA methylation was absent in CVL125. A potential candidate might be the Cvi allele of the *NUCLEAR RNA POLYMERASE D1B* (*NRPD1B*) gene, which contains several SNPs that might affect its function (1001genomes.org). *NRPD1B* was recently identified as a major *trans*-acting locus affecting DNA methylation in *Arabidopsis*[8] (all other major *trans*-acting loci identified in that study are of Ler origin in both, CVL39 and CVL125, Supplementary Fig. 2). It may also explain the preferential loss of DNA methylation in the CHG context in CVL125 compared to the gain of DNA methylation in the CHG context in CVL39 (Fig. 2e). Although speculative, these observations suggest that the epigenotype may take several generations to align with its new genetic background: in case of the selected individuals of CVL125 that still carried the methylated *At2g06002* allele, the allele persisted for 8 to 16 generations since a putative *trans*-acting locus enforcing methylation was lost (8 generations correspond to the 1 generation bulk-up, 5 generations under selection, and 2 generations in the common environment, the other 8 generations are possible if the locus enforcing methylation in *trans* was lost early while generating the RILs in ref. [18]).

If the epigenetic variation observed in our study was, at least to a certain extent, a consequence of the initial hybridization between Cvi and Ler and a delayed alignment of the epigenotype

with the genotype, it would have important implications for future studies. Even though such variation may be functionally relevant and could buffer phenotypic changes over generations, it should clearly be separated from spontaneous, random epimutations or environmentally induced epivariation. However, this is only possible if data from several generations of ancestors are available. Furthermore, it would suggest that ecologically and evolutionary relevant epigenetic variation may more frequently contribute to adaptation in genetically diverse and outcrossing species than in self-compatible or asexually reproducing species. This may be unexpected because it is frequently argued that epigenetic variation may evolutionary be more important in populations with low genetic diversity and asexually reproducing species (e.g., refs [16,49]). However, although epigenetic variation may be more frequent in genetically diverse species, genetic diversity is much higher as well. Hence, the relative importance of epigenetic variation may still be higher in populations with low genetic diversity and asexually reproducing species.

In conclusion, although the origin of selected epialleles is still unclear, our studies have shown that selection can lead to novel phenotypes that are stably inherited for 2–3 generations, and which are highly unlikely to be caused by the small number of SNPs observed. Thus, we provide evidence that epigenetic variation is subject to selection and can play a role in fast adaptive responses. However, the relative extent to which genetic and epigenetic variation contribute to plant adaptation remains to be elucidated and likely depends on the reproductive mode of the investigated species.

## Methods

**Plant material and growth conditions**. The original selection experiment from which the plants used in this study were derived, was fully described elsewhere[17]. In brief, the experiment started with a population of 19 genotypes, i.e., 17 RILs and their two parental accessions, Cape Verde Island (Cvi) and Landsberg *erecta* (Ler). RILs were established and characterized previously[18]. Seed for the original selection experiment was obtained through NASC and propagated for one generation in a standardized greenhouse environment to amplify seed stocks and to reduce potentially confounding maternal effects. Offspring of this "original population" was then grown for five generations in a selective environment simulating a fragmented habitat. After five generations, genetic diversity was strongly reduced, and only two genotypes (CVL39 and CVL125) dominated the populations grown in dynamic landscapes[17]. For the present study, seeds were taken from the original founder population ("ancestral", D0) and from populations of three dynamic landscapes, i.e., replicated, independent selection experiments ("selected", D1, D5, D6). Plants were then grown for three generations (A1/S1, A2/S2, A3/S3) together in a randomized matrix in a controlled environment (Supplementary Fig. 1). Seeds were sown on agar plates and stratified at 4 °C for three days. To identify CVL39 and CVL125 individuals among all other possible genotypes, plants of the first generation (A1/S1) were genotyped using nine Indel/SSLP markers[17]. For genotyping, one cotyledon was harvested and frozen in liquid nitrogen ten days after sowing. DNA was extracted in a 96-well format using Edwards DNA extraction buffer[50]. After 12 days, seedlings were transferred to pots (two individuals of the same RIL per pot) with ED73 soil (Einheitserde, Germany), and grown under long-day conditions (23 °C, 16 h light, 8 h dark). Positions of pots were randomized. Seeds of individual plants were harvested separately after 12 weeks. Collected seeds from individuals were propagated in the next generation. The same growth conditions were applied to all three generations. Phenotypes, methylomes, and transcriptomes were measured on different individuals from a given population. Additionally, after the material for methylome profiling had been harvested during the second generation, the populations got infested with thrips and could not be used for further studies. Therefore, we started with new seeds from the first generation for all other experiments (phenotype, transcriptome, genome resequencing). Number of rosette leaves at bolting was recorded around day 20–25 (i.e., the day of bolting). The total number of seed pods, branches, and stems were measured at day 64–73. Shoots growing from the rosette were classified as main stems (with several side branches) or stems (with one to three side branches). Branches were defined as inflorescences grown from any stem or branch. Shoots growing from the rosette without branches were classified as branches as well. For the methylome and transcriptome, above ground parts of individual plants were harvested at day 25, placed in an Eppendorf tube containing glass beads, and flash frozen in liquid nitrogen.

**Analysis of phenotype data**. Variation in phenotypic traits was analyzed with a general linear model in R[51], according to a crossed factorial design with the three explanatory factors GEN (generation: second/third), RIL (recombinant inbred lines: CVL39/CVL125), and POP (population of origin: D0/D1/D5/D6), and all

interactions between them. Because the plants of the second and third generation were grown at two different time points, GEN was treated as a blocking factor (i.e., its contribution could not be separated from the two-time blocks), and was therefore not further interpreted. POP was divided into a 1-degree-of-freedom contrast evoPOP (ancestral population, D0, versus selected populations, (D1 + D5 + D6)/3) and remPOP (remaining differences among the three selected populations D1/D5/D6; Supplementary Data 1).

**Illumina whole-genome bisulfite sequencing (BS-Seq)**. For whole-genome BS-Seq, genomic DNA was extracted from flash frozen plant material using the DNeasy Plant Mini Kit (Qiagen, Switzerland). 150–500 ng of genomic DNA were physically sheared to an average sequencing library insert size of 300 bp (10% duty cycle, intensity 4200 cycles per burst, 40 s) in 120 μl of 1 × Tris-HCl/EDTA using the Covaris S2 system (Covaris, USA). Sheared DNA was purified and size selected using 1.8 × the volume of Agencourt AMPure XP magnetic beads (Beckman Coulter, Germany), following the standard procedure including two 80% ethanol washes. Dried beads were re-suspended in 62.5 μl of Illumina re-suspension buffer (Illumina, USA), which yielded 60 μl of purified DNA. Sequencing libraries were prepared according to the "Low Throughput" protocol of the TruSeq DNA Sample Preparation v2 Guide, omitting the library amplification step. Adapter-ligated DNA was eluted in 25 μl of Illumina re-suspension buffer and subjected to bisulfite conversion using the Epitect Bisulfite Kit (Qiagen, Switzerland). A standard reaction mix consisting of 15 μl DNA protect buffer was used for bisulfite conversion in a thermal cycler (5 min at 99 °C, 25 min at 60 °C, 5 min at 99 °C, 85 min at 60 °C, 5 min at 99 °C, 175 min at 60 °C). After the incubation period, bisulfite converted DNA was purified using the Epitect Bisulfite protocol for DNA isolated from FFPE tissue including carrier DNA. DNA was then eluted in 25 μl of elution buffer (Qiagen, Switzerland). The bisulfite converted sequencing library was enriched with Pfu Turbo Cx Hotstart DNA polymerase (Agilent Technologies, Switzerland), using a protocol adapted from Feng et al.[52] Fifteen PCR amplification cycles were carried out. Amplified libraries were purified using the Qiaquick PCR purification kit (Qiagen, Switzerland) and eluted in 30 μl of Illumina re-suspension buffer. Libraries were validated and quantified for sequencing on the Agilent TapeStation using a High Sensitivity D1K Screen Tape (Agilent Technologies, Switzerland). Single-indexed libraries were paired-end sequenced on the Illumina HiSeq 2500 system (Illumina, USA).

**Reference genomes for the RILs**. To create recombinant genomes suitable for the alignment of BS-Seq reads, we used reads of a pilot BS-Seq experiment (low-coverage SOLiD data, GEO accession number GSE36845). L*er* and Cvi reference sequences were constructed using the Col-0 reference genome (TAIR10) and SNP annotation available on TAIR. Reads were then aligned to these two parental reference genomes with SOCS[53] (version 2.1), allowing for up to 4 mismatches in addition to tolerating T-to-C and A-to-G substitutions. From the variation in mismatch data between alignments of parental lines, we could estimate recombination points for CVL125 and CVL39 down to around 2.8 kb resolution and construct the reference sequences for the two RILs. We noticed a region on chromosome 2 of approximately 480 kb, which was heterozygous for L*er*/Cvi in the CVL125 population. We genotyped the CVL125 individuals used in the phenotypic assessment and found that the relative contribution of individuals of each genotype was the same in the original CVL125 NASC seed and the selected populations, indicating that the heterozygosity in this region was not under selection. The recombinant reference genomes are available upon request.

**Alignment of BS-Seq reads**. The totally 1,459,122,191 reads generated by Illumina BS-Seq were quality-checked with FastQC (bioinformatics.babraham.ac.uk/projects/fastqc). Following removal of adaptor sequences and low-quality reads (Trimmomatic[54], version 0.30 with the parameters LEADING:5 TRAILING:5 SLIDINGWINDOW:5:15 AVGQUAL:20 HEADCROP:2 MINLEN:50), reads were aligned to recombinant genomes using Bismark[55] v0.10.0 in conjunction with Bowtie2[56] (version 2.2.4), with the following parameters specified—score-min L,0, −0.2 (i.e. allowing for up to three mismatches in addition to tolerating T-to-C and A-to-G substitutions). Clonal reads with identical sequences resulting from possible over-amplification during sample preparation were removed with Picard tools (version 1.128, sourceforge.net/projects/picard). Only reads aligning uniquely to the reference genome were used for subsequent analyses. The bisulfite conversion rate was on average 99.7%, and in all samples higher than 99.3%, as assessed from the unmethylated chloroplast genome[28,57]. Methylated and unmethylated read counts for all cytosines across the genome in the CG, CHG, and CHH context were obtained from Bismark bisulfite census files. Cytosines with an average coverage below 5 and above 100 across each genotype were removed to avoid a potential bias originating from low coverage or from poorly annotated sequences[58]. The samples had on average a genome coverage of 36.2 after filtering (Supplementary Data 2), which corresponds to the "gold-standard" per sample coverage in Ziller et al.[59], and is well above most previous studies on DNA methylation in plants (for example on average 12.6×[7], 16×[60], 20–27×[21], and 6–25×[4]).

**Mean pairwise distances (MPD)**. The MPD in DNA methylation patterns between the individuals of a given population reflects the epigenetic diversity

within the population. Pairwise distances between two individuals were calculated for a given context and chromosome as the average methylation level differences across all cytosines. It has been shown that MPDs are independent of the number of individuals[20,61].

**Differences to original accessions**. To show the similarity of DNA methylation patterns to the original accessions along the genome (Supplementary Fig. 4), we calculated the difference between the average pairwise distance of ancestral and selected individuals to Cvi and L*er*. Pairwise distances between an ancestral or a selected individual and one of the original accessions were calculated for a given context and genomic bin of 10 kb size as the average methylation level differences across all cytosines within the bin (data for Cvi and L*er*-1 was taken from neomorph.salk.edu/1001_epigenomes.html).

**Determination of DMCs**. Given the solid number of individual replication (four and eight individuals per selected and ancestral population, respectively, for each genotype) and the high coverage per individuum (see above), it was possible to test each single cytosine for differential methylation instead of summarizing entire genomic regions (i.e., differentially methylated regions, DMRs). To test whether methylation at single cytosines was selected, we modeled the methylation level in percentage as a response to the selection scenario with a linear model similar to the one described above for the analysis of phenotypic traits with POP (population of origin: D0/D1/D5/D6) as an explanatory factor. Because the two genotypes (CVL39/CVL125) did not share all cytosines and the main focus lied on the identification of selection of epigenetic variation, each genotype was analyzed separately. POP was divided into a 1-degree-of-freedom contrast evoPOP (ancestral population, D0, versus selected populations, (D1 + D5 + D6)/3) and remPOP (remaining differences among the three selected populations D1/D5/D6). *P*-values for evoPOP and remPOP were adjusted for multiple testing using the approach proposed by Storey[62] (*Q*-values). A cytosine was defined as differentially methylated (DMC) if only evoPOP, but not remPOP, was significant (*Q* < 0.05, Supplementary Data 3, 4).

**Genomic sequencing of CVL39 ancestral and selected lines**. The Low Sample (LS) protocol and reagents as described in the Illumina TruSeq DNA Sample Preparation Guide p55ff (Illumina Part #15026486 Rev. C July 2012) was followed to prepare paired-end genomic sequencing libraries from 500–1000 ng genomic DNA. Sequencing was performed on an Illumina Highseq 2000 instrument. Genomic libraries of 2 × CVL39A3 ancestral and 7 × CVL39S3 (from 2 × D1, 3 × D5, 2 × D6 replicate landscapes) selected lines were sequenced with 100-bp paired-end reads. For image analysis and base calling, we used the Illumina RTA and CASAVA software version 1.8.2. Reads were mapped to the recombinant CVL39 genome reference sequence using Bowtie2[56] with default settings. SNPs were determined by using the unified genotyper of the Genome Analysis Toolkit 2.1.6[63].

**Identification of SNPs**. To identify SNP positions between the two ancestral and seven selected CVL39 samples we used tools implemented in PoPoolation2[64] (version 1.201). Briefly, the two major alleles for each SNP position were identified and Fisher's exact test was applied to test whether any differences in allele frequencies between ancestral and selected lines were significant (*P* < 0.05). A minimum read coverage of 20× and a maximum coverage of 200× was set to eliminate regions that had a too low coverage for SNP identification and to rule out artefacts as a consequence of incomplete annotation of repetitive elements in the reference genome. To qualify as a SNP, at least four selected lines (> 50%) had to contain different alleles compared to at least one of the two ancestral lines. Five homozygous SNPs that were significant between both ancestral and all seven selected lines were subjected to Sanger sequencing using at least 19 additional ancestral and 12 additional selected lines (from 4 × D1, 4 × D5, 4 × D6 replicate landscapes). Sequences between 301 and 393 bp length (dependent on the SNP) flanking the SNP locus were first amplified using standard reagents (Sigma-Aldrich, Switzerland) and purified using NucleoSpin Extract II columns (Macherey-Nagel, Switzerland). PCR for sequencing was carried out as follows: reaction mix of 2 μl DNA template (20-100 ng), 5 μM forward primer, 2.5 μl sequencing buffer (5×), 0.85 μl big dye terminator and 5.15 μl water; PCR: 1 × 94 °C 2 min, 60 × (94 °C 10 s, 50 °C 5 s, 60 °C 3 min), 1 × 4 °C 15 min. Millipore MultiScreen plates (Millipore, Switzerland) with Sephadex G-50 Superfine (Amersham Biosciences, Switzerland) were used for dye terminator removal following standard protocols and conditions. Sequencing was performed on the Applied Biosystems 3730 DNA Analyser. Sequences were visualized and analyzed in the CLC Main Workbench 6.5 (CLC bio, Denmark). SNPs that showed two peaks in the SNP position of similar height in the trace data were termed heterozygous.

**Identification of transposable element (TE) insertions**. Paired-end Illumina sequences (R1 and R2) were aligned separately to annotated TEs of *Arabidopsis* as well as to the reference genome (TAIR10). First, we identified sequence pairs where one of the pairs mapped within an annotated TE sequence. Second, since paired reads share the same sequence identifier, genomic positions of the second pair that were located outside of TE regions were identified based on genome mapping. A novel TE insertion had to fulfill several criteria: at least 30 paired-end reads were

required per event and the distance of new insertions to the Col-0 TE reference location had to be a least 4 kb. Genomic positions of TEs were then compared between the two ancestral and the seven selected lines to identify novel TE insertions in the selected genomes. Novel TE insertions were scored if at least four selected lines (> 50%) had TE insertions that were not present in both of the ancestral lines.

**Mapping of genomic positions to local genetic context**. Genomic positions (e.g., DMCs) were mapped to their local feature context using the TAIR10 annotation. Regions that lacked annotations were defined as intergenic. Genes were further broken down into introns, exons, 5′-UTR, and 3′-UTR. For methylation statistics (Fig. 2f), annotations were given equal priorities and their score was increased by the fraction of the number of features that mapped to the DMC. For functional analysis, all annotations were used. The direction of methylation change for each gene containing DMCs was calculated from the average change across all DMCs in the gene. To assess the number of DMCs present in both genotypes, empirical distributions were calculated using 10,000 random sets of positions from a list of all tested Cs (for each genotype, the number of randomly sampled Cs was equal to the number of DMCs).

**Association of DMCs with 24-nt siRNA and transposons**. Publicly available siRNA datasets[28,65,66] were used to generate a list of 24-nt target regions. 24-nt siRNAs closer than 10 bp to each other were merged into a single target region. Genomic positions (e.g., DMCs) were then mapped to these target regions. To test for co-occurrence of DMCs and 24-nt-siRNA target regions, and to assess the distance between DMCs and the closest 24-nt siRNA target regions, we obtained empirical distributions of co-occurrences and distances using 500 random sets of positions drawn from a list of all tested Cs. To avoid sequence context bias, the positions were drawn separately for a given genotype and sequence context (e.g., 1342 random Cs in the CHG context for CVL39 with 1342 CHG-DMCs). The method described here was also used to test for association of DMCs with transposons.

**Metagenes**. Generic gene models were constructed from protein-coding genes (TAIR10) of at least 100 bp in size, excluding genes within a distance of 1 kb of the chromosome ends. The distribution of DMCs along the generic gene body was obtained by dividing each protein-coding gene into 100 bins. For the 1 kb flanking regions, the average DMC coverage was directly obtained and smoothened. Plots were done in Python (version 2.7.3) using numpy (version 1.6.1, numpy.scipy.org) and matplotlib (version 1.1.1rc, matplotlib.sourceforge.net).

**Boxplots**. All boxplots were produced in R[51] with the default boxplot function. The bottom and top of the boxes correspond to the lower and upper quartiles and the center line marks the median. Whiskers extend to the lowest/highest values unless these values are lower/higher than the first/third quartile minus/plus 1.5 times the inner quartile range (IQR), which equals to the third minus the first quartile.

**GO enrichment**. To functionally characterize the genes associated with DMCs, we tested for enrichment of GO terms over a range of different thresholds for the number of DMCs per gene (1, 2,… 20 in steps of 2) and the average change in methylation levels (0,… 50 in steps of 5). Within a given combination, we used topGO 2.20[67] in conjunction with the GO annotation available through biomaRt[68,69]. Analysis was based on gene counts (protein-coding genes with DMCs compared to all annotated protein-coding genes) using the "weight" algorithm with Fisher's exact test (both implemented in topGO). A term was identified as significant within a given parameter combination if the P-value was below 0.05. To extract the set of GO terms exhibiting a robust enrichment, only terms found to be significant in at least 50 (out of 121) parameter combinations in CVL39 and/or CVL125 were considered (Supplementary Data 9).

**Analysis of gene expression**. Gene expression differences between four ancestral A2 and four selected S2 individuals of each genotype were assessed using AGRONOMICS1 Tiling Array Genechips (Affymetrix and Functional Genomics Centre Zurich (FGCZ)), which cover 29,920 TAIR9 gene models including all the ATH1 Affymetrix Genechip sense probes. Approximately half of the frozen whole plant tissue from 25-day old plants was used for RNA extractions. Total RNA was isolated with the TRIzol Reagent (Invitrogen, USA) and treated with DNAse (Applied Biosystems, AM1906) to remove any contaminating DNA. 2 μg of RNA were reverse transcribed using standard Invitrogen reagents and protocols. cDNA samples were subjected to RNAse H digestion to remove any remaining RNA/DNA hybrid complexes (Invitrogen, USA). The quality of RNA (integrity and purity) and cDNA were assessed with the 2100 Bioanalyzer pico Chip (Agilent Technologies, USA). cDNA was quantified with the NanoDrop 1000 Spectrophotometer (Thermo Fischer Scientific, USA) and diluted to 100 ng per μl with sterile water. cDNA was hybridized to AGRONOMICS1 Chips (FGCZ) using standardized assays and reagents for Affymetrix GeneChip Technology. Expression signals were normalized using the Robust Multichip Average (RMA) approach[70], defined in custom functions for analyzing AGRONOMICS1 Chip data (www.agron-omics.eu/

index.php/resource_center/tiling-array/tools-and-protocols), implemented in R and bioconductor, which also requires the aroma.affymetrix package[71,72]. The R package limma[73] was used to fit a linear model to the normalized expression data for each gene across ancestral (A2) and selected samples (S2), and to calculate the differences in gene expression. P-values were adjusted for multiple testing to reflect false discovery rates (FDRs). A gene was considered to be differentially expressed if the log2 fold-change was at least 1 and the FDR was below 0.05.

**Analysis of publicly available data**. Preprocessed DNA methylation and gene expression data from Schmitz et al.[31] were retrieved from GEO (GSE43857 and GSE43858, only accessions for which both transcriptome and methylome data were available). DNA methylation levels at DMC positions at the locus At2g06002 and its flanking regions (1 kb) were extracted and averaged. The average DNA methylation level separated the 121 accessions into two groups with either high or low DNA methylation levels. Differences in gene expression and phenotypic traits (from Atwell et al.[32]) between these two groups were analyzed with a two-sided t-test (only 54 accessions with phenotypic traits, Supplementary Data 14). The list of traits can be accessed under archive.gramene.org/db/diversity/diversity_view).

**Droplet digital PCR**. To extract RNA for the ddPCR, tissues were harvested using the same methods as for DNA extraction for BS-Seq. Total RNA was isolated with the RNeasy Plant Mini Kit (Qiagen, Switzerland) according to the manufacturer's protocol (RLC buffer, elution in 30 μl). DNase treatment was done with the Turbo DNA-free Kit (Invitrogen, USA) according to the manufacturer's protocol (3 μl Turbo DNase, incubated at 37 μC for 30 min). 4 μg of RNA were reverse transcribed using standard Invitrogen reagents and protocols (RT+). For each sample, an additional mock reaction (RT−) without the addition of SuperScript II Reverse Transcriptase was carried out to control for genomic DNA contamination during ddPCR. We could not detect notable genomic contamination, see also Supplementary Data 12.

As far as possible, we used assays already described in the literature. Primers were available for two reference genes (PP2A and UBC9[74]), FRI[75], and FT[76]. Primer sequences for the remaining genes were designed using the CLC Main Workbench software. All primer sequences used are listed in Supplementary Table 2. All primers were tested and validated for optimal concentration, and primer efficiency was assessed for the three newly designed primer pairs. Primers were tested in a 7500 Applied Biosystem Fast quantitative Real-Time PCR System and later validated on a QX200 Droplet Digital PCR System (Bio-Rad, USA) for the ddPCR assay. Reactions for qPCR were performed in total volumes of 20 μl containing 10 μl 2X SYBR-green Supermix (SsoAdvanced Universal SYBR). For the ddPCR analysis, individual PCR reactions were performed in a total volume of 25 μl, using 1 × ddPCRTM EvaGreen Supermix, with droplets generated according to manufacturer's recommendations. Reading of the PCR-amplified droplets was carried out by the QX200 Droplet Reader and analysed by the QuantaSoftTM Software (v1.4, Bio-Rad).

Raw data are provided in Supplementary Data 12. To compare the expression of the genes between the populations, we calculated log2 ratios between the test genes and the geometric mean of the reference genes[77,78]. RT+ counts of the test genes and the reference genes were first log2$(x + 1)$ transformed, and the value of the reference genes was then subtracted from the value of the test gene. Populations were compared to each other with two-sided t-tests. For each gene, P-values were adjusted for multiple testing. Adjusted P-values (FDR) below 0.05 were considered to be significant (significance letters in Supplementary Fig. 3). To compare the selected populations with the ancestral population, we used a linear model similar to the ones described above for the analysis of phenotypic traits and DMCs. However, only the results of the contrasts evoPOP are shown in Supplementary Fig. 3.

## Data availability
All relevant data generated in this study were deposited at the NCBI Gene Expression Omnibus (GEO) and the NCBI Sequence Read Archive (SRA), and are available through accession number GSE36844 (microarrays), GSE36845 (low-coverage pilot WGBS), GSE47490 (genome sequencing), and SRP059356 (WGBS data).

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

## Acknowledgements

We are grateful to S. Fakheran and C. Paul-Victor (University of Zurich) for providing seeds of the ancestral and selected populations from which CVL39 and CVL125 were identified. We thank S. Tierling (Saarland University) for providing bisulfite conversion protocols, and R. Schlapbach (Functional Genomics Centre Zurich, University of Zurich) and T. Leeb (NGS Platform, University of Bern) for providing access to experimental facilities and human resources for Next Generation Sequencing and GeneChip experiments. This work was supported by the University of Zurich, a Syngenta-Fellowship Project of the Zurich-Basel Plant Science Center to U.G., B.S. and L.A.T., a pilot project grant from the University Research Priority Program Functional Genomics/Systems Biology to C.H. and U.G., and grants from the Swiss National Science Foundation and the European Research Council to U.G.

## Author contributions

U.G. conceived the project; U.G., B.S., and L.A.T. raised funding and supervised the project; C.H. designed and performed the experiments with assistance from D.G., D.C.S., S.A., V.G., and C.A.; M.W.S., C.H., D.C.S., R.B., V.G., B.S., L.A.T., and U.G. analyzed and interpreted the data; M.W.S., C.H., and U.G. wrote the manuscript with assistance from all authors.

## Additional information

**Competing interests:** The authors declare no competing interests.

