## [Peer Review File · Nature Communications]

Reviewers' Comments:

Reviewer #1:

Remarks to the Author:

The manuscript by Schmid et al takes advantage of a multi-generational experiment on recombinant inbred lines (RILs) of a cross between CVI and Landsberg that were grown in so-called "Dynamic Landscapes" for 5 generations. The design may have a lot to offer for the question of whether DNA methylation contributes to adaptive change, but the manuscript suffers from a lack of information on exactly what it can offer- e.g. we don't actually know the details of methylation of individuals and how that is specifically correlated to the details of individual gene expression and phenotypic variation. In fact it appears that the DNA methylation was measured on one set of plants and the gene expression on another. This type of design flaw ignores the fact that DNA methylation is extremely dynamic and can have specific impacts on gene expression and phenotypes that must be measured on the same individual. Considering that the point of the study is to link DNA methylation specifically to adaptation, it's not clear that this can be done with the design as it is described.

The authors also claim that "Thus, epigenetic variation likely contributes to adaptation through the same mechanism as genetics and not through inheritance of environmentally induced epimutations." (lines 110-111) but don't provide a compelling reason. They claim that the patterns of DMCs could be explained by "selection of standing epigenetic variation instead of the selection of de novo acquired epimutations" (106-107 and 127-129) which seems problematic for two reasons: the most obvious is that regardless of whether the epialleles are already present in the population or induced, the pattern shows a correlation with fitness so DNA methylation is associated with adaptation. The authors seem to be arguing against induced epi-alleles for some undescribed reason. The second problem with this reasoning is that the authors have the data to examine whether the methylation was present in the ancestral population. It's not clear why they have argued that the variation was there already when they just spent several paragraphs describing how the patterns are different in the selected populations. If that variation was there for many of the loci, but perhaps in lower frequency, they should be able to find it for some portion of the loci in the ancestral plants that were sequenced. Otherwise, there is no reason to believe that the methylation patterns behave the same as genetic loci. Overall reduction of DNA methylation does not mean necessarily that what's left is a subset of what was there or preclude novel patterns (lines 201-203).

From the first paragraph:

The authors should acknowledge that heritable variation may come in more than two so-called "flavors" considering that they have only just investigated 2.

Line 33- This description of the experiment is confusing. The phrase "reveal the outcome of selection" implies that the source of selection is understood (eg drought, heat, competition etc), but the authors don't describe the selective environment. It could be better to more generally describe "response to novel natural field environments" or something along those lines. It's also not clear here or elsewhere what is meant by "only two genotypes (CVL39 and CVL125) dominated the populations". Were these two equally represented across all of the populations? Or did one dominate some and the second dominate the others? This type of disparity would indicate that perhaps the two were differentiated by habitat type (biotic or abiotic). We haven't time to go back to the original study, so this important information should be provided.

Line 47 resequencing individuals from 1 of the genotypes will not tell you necessarily that the same thing happens in all individuals of that genotype or if the findings would be the same for the other genotype. e.g. the single random mutation in MEE57 in one of the mutation accumulation lines changed genome-wide methylation of otherwise Col wild type. Incidentally, didn't you also sequence 24 individuals (line 66) that would shed light on this question?

lines 54-61 These arguments about phenotypes and fitness are quite convincing and a real strength of the manuscript.

line 89 Its unclear what this sentence means: "Only 20/23 terms were significantly enriched in at least 50 (out of 121) parameter combinations in CVL39 and CVL125..."

line 131-141 This description and notation are very confusing as written.

line 137 which plants are defective in DNA methylation pathways?

line 143 there's no reason to expect a global pattern of DNA methylation association with expression. We know that a lot of methylation is stochastic and seemingly non-functional. The point is to find the few that might be functional.

lines 145-165 This analysis of methylation and expression of At2g06002 is definitely of interest and more of this would be welcome. However, given that the methylation was measured on different plants than the expression, its not clear how accurate these findings might be?

lines 166-175 Again, very interesting discussion, but what are the actual patterns of DMCs? Its difficult to decipher which expression patterns were associated with changes in methylation. The authors only suggest that FT is not associated with DMCs.

lines 186-190 The argument made here that " Although we cannot fully exclude the possibility that segregating alleles partly contributed to the observed phenotypic changes or differences in methylation patterns, it is very unlikely that they underlied these changes because none of the affected genes (or genes near an intergenic SNP) have a known role in processes related to flowering time, plant stature, or DNA methylation. " is not so convincing considered how little we actually know about how the genome functions and translates into phenotype.

Reviewer #2:

Remarks to the Author:

The manuscript from Schmid and colleagues tackles an interesting question: Whether epigenetic variation can be acted upon by natural selection and thereby contribute to adaptation. The authors make good use of existing populations of recombinant inbred lines that had been previously subjected to selection in a simulated fragmented habitat. The goal of the experimental design was to make good seed dispersal essential and reduce competition between genotypes. The authors examined the ancestral population and three selected, descendant populations. The ancestral population contained nineteen genotypes, and the descendant populations had been subjected to five generations of selection. The prior study found that two genotypes dominated the descendant populations. This is summarised in the submitted manuscript (and clearly, appropriately attributed), but is slightly difficult to follow. The authors might consider moving Fig 1A to a separate figure and expanding it - it took me some time to understand that their new analyses focused on individuals of the two dominant genotypes but that these were not the only genotypes in the populations.

The authors present reasonably persuasive evidence that epigenetic diversity is reduced in the genotypes that dominate the descendent populations relative to individuals of those genotypes, correlating with phenotypic differences (Fig 1B, C). This result is key to all subsequent data interpretation. I would consequently like the authors to comment on the difference in the number of ancestral and dependent individuals used to calculate the mean pairwise distance between DNA methylomes. Only four ancestral individuals were used, compared with 8 per genotype (16 total) descendants. I am unfamiliar with the mean pairwise distance statistic; how does it take account for differing numbers of individuals and what possibility is there that the relative diversities are

artefacts of this?

The authors demonstrate that their results indicate selection has acted upon epigenomic variation that existed in the ancestral population, rather than on environmentally induced epigenomic variation. They go on to focus upon a candidate differentially methylated region around At2g06002 that may impact flowering time, using phenotypic and epigenomic data from 54 Arabidopsis accessions to further assess the region's significance. This approach is convincing. Can the authors please explain, though, whether the two variant DNA methylation states found in the descendent populations were detected in any ancestral individuals? Also, how do they reconcile the fact that in one dominant descendent genotype expression of At2g06002 was increased and in the other dominant descendent genotype it was increased, but both descendent genotypes have delayed flowering? I think their argument needs some rephrasing in that section.

Reviewer #3:

Remarks to the Author:

This paper argues that selection on epigenetic variation has contributed to phenotypic change in two recombinant-inbred line populations. I first heard these results presented in a meetings years ago, and it would be great if they were finally published. Presumably the authors have suffered from the tendency in this field of declaring results as either trivial and well-known or potentially very interesting but unproven. My personal opinion is that these result are both interesting and novel, that the level of proof is good enough, but that we still don't know what is going on. I have a few concerns, though...

First, I would moderate claims about the importance of selection on epigenetic variation. You have not really demonstrated (as stated in the Abstract) that "epigenetic plays a role in fast adaptive responses", but rather that it CAN play a role. Similarly, on line 25, you mention "a considerable contribution of induced epialleles". I'm sure there are other such statements, and I don't see how they can be supported without explicitly comparing the epigenetic to the genetic contribution. You are actually in a position to do so: in your selection experiment, the several traits changed dramatically because of selection among RILs (only two survived). Then you also show that the same traits changed within RILs. Which contributed most to the change in phenotype?

Second, I'm confused by your arguments about standing epigenetic variation. Surely the original experiments were bulked from single-seed descent RILs? Does your explanation work, quantitatively?

Third, what would account for the big difference between the two RILs? It may be worth considering our recent results (Dubin et al. 2015, Kawakatsu et al. 2016) demonstrating the existence of major trans-acting variation affecting methylation. Cvi, in particular, is known to have very different methylation levels, and the alleles responsible for this must be segregating in the Cvi-Ler RIL population. It is thus possible that epimutation rates differ between lines, and also that the entire genome is not at steady state w.r.t. the genetic background.

Fourth, I'm very confused by what you mean by a DMC. Your premise is that methylation states are inherited. Well, then the methylation states should be either 0 or 1 (or possibly 1/2). Like a SNP. In each replicate. Looking at your supplementary tables 3 and 4, you compute averages of methylation percentages, some of which are certainly not 0 or 1. This is not internally consistent. Especially, how can you argue that CHG and CHH sites are part of epigenetic selection, when the methylation fractions per site are far from 0 or 1...? I should emphasize that I don't know what is going on either, but it does seem that your data rule out methylation generally acting as a "fifth base", which is kind of what you are arguing it does... This is must be addressed somehow, if only be acknowledging that there is a problem... More constructively, can't you explicitly test for methylation variants that appear to behave like SNPs?

Happy to chat more in person,

Magnus Nordborg

Reviewer #4:

Remarks to the Author:

This manuscript by Schmid et al provides evidence that epigenetic variation can be subject to natural selection and lead to adaptive phenotypic changes in a changing environment. The study is based on a previously published selection experiment by the same authors (Fakheran et al PNAS 2010), where starting from recombinant inbred lines (RILs) natural selection in a simulated disturbed habitat over five generations was applied, and phenotypic differences were recovered (growth, flowering). The present study aims to identify the underlying molecular variation that is responsible for these phenotypic differences. Since the starting material (the RILs) was supposedly genetically identical, the authors focused on variation at the epigenetic/ DNA methylation level. By analyzing two independently selected lines derived from two different RILs, the authors identify through methylome and transcriptome analysis a large number of differential DNA methylation regions (DMR) that may account for the phenotypic differences. In one of the lines they identify a DMR that is associated with differential expression of a non-coding RNA and at the population level is associated with altered flowering time at 10°C. However, this was not affected in the other line, where instead they find changed expression and methylation in a number of circadian clock and flowering time genes. This suggests that the molecular basis of the phenotypic changes may not be the same in the two lines. The authors conclude that the DMRs were selected from standing variation in the parental RILs (residual heterozygosity for DMRs).

The question of whether epigenetic variation can be selected for in natural variation is highly timely and of high interest in the research community. The study is conducted elegantly and thoroughly and the manuscript reads well. However, the underlying hypothesis is that there is no genetic variation left in the RILs and thus any phenotypic changes must be caused by epigenetic changes, - the authors themselves provide evidence that this is not the case. In a genome resequencing effort that covers only one of the two selection lines under study, they identify 14 SNPs, most of which were present as standing variation also in the parental RIL line. This finding is brushed away with the statement that there is no known function for the affected genes in the relevant processes. How can the authors know this? It remains possible that this genetic variation at least contributes to the phenotypically selected traits. The authors should provide additional evidence to strengthen their case (genetic vs. epigenetic basis) or phrase their claims more carefully. For example, l. 17 of the abstract is not correct and needs to be rephrased.

The design of the -omes-experiments is not symmetrical. While for the methylomes extensive data from 3 replicate experiments of both selected lines are collected (CVL39, 125; D1, D5, D6), the transcriptome analysis is performed only for D1 of CVL39 and 125. The genome resequencing is only done for CVL39; D1, D5, D6). As the most important criterion in the end appears to be differential gene expression, wouldn't it make more sense to analyze also D5 and D6 and then look for common differentially expressed genes? Interpretation of the data should also bear in mind that the transcriptome data give only a snapshot of one particular developmental stage/time of day.

Minor comments:

l. 148: "was higher than..." - how much higher?

l. 158: flowering time at 10°C appears to be a rather exotic condition. Surely, Ref. 24 or other references must provide flowering time results under various conditions (including more typical ones). Is this the only one where an association was found or was this one selected for a specific reason? This should be stated/ explained more clearly.

l. 163: Is the expression of FIP1 affected?

Fig. 1B: Do these data stem from the 2nd or 3rd generation?

Fig. 2A: It took me a while to understand the Figure. Maybe the relevant part of the Figure could be highlighted to make it more clear?

Detailed Response to the Reviewers' Comments

We would like to thank all reviewers for their interest in our work and the efforts they put into providing constructive criticisms and insightful comments. Their comments have helped us to improve the manuscript and express our ideas more clearly. Below we provide a point-by-point response to all comments. To facilitate reading, the reviewers' comments are in italics, while our responses are in roman and indicated by **RESPONSE**.

Reviewer 1

The manuscript by Schmid et al takes advantage of a multi-generational experiment on recombinant inbred lines (RILs) of a cross between CVI and Landsberg that were grown in so-called "Dynamic Landscapes" for 5 generations. The design may have a lot to offer for the question of whether DNA methylation contributes to adaptive change, but the manuscript suffers from a lack of information on exactly what it can offer- e.g. we don't actually know the details of methylation of individuals and how that is specifically correlated to the details of individual gene expression and phenotypic variation. In fact it appears that the DNA methylation was measured on one set of plants and the gene expression on another. This type of design flaw ignores the fact that DNA methylation is extremely dynamic and can have specific impacts on gene expression and phenotypes that must be measured on the same individual. Considering that the point of the study is to link DNA methylation specifically to adaptation, its not clear that this can be done with the design as it is described.

RESPONSE: Indeed, DNA methylation and gene expression were measured on two different subsets of the phenotyped plants. In addition, gene expression of selected plants was only measured in individuals from one of the replicated landscapes. Thus, we indeed may have missed correlations that are specific to individuals (i.e., exist in some, but not in others). We could also have missed some correlations between DNA methylation and gene expression (and phenotype) because the transcriptome provides only a single snapshot in the life cycle (e.g., some correlations may be apparent only at other developmental stages). While these are caveats of our analysis and we mention them as possible reasons for the absence of correlation in the revised version of the manuscript, we do not think that it they constitute a serious concern. This is because selection of correlations specific to individuals can only be weak. For epialleles to be selected, they must be translated faithfully into an advantageous phenotype. If this is achieved through regulation of gene expression, then the correlation must be robust as well. Unless a correlation was missed due to the snapshot nature of our transcriptome analysis, our study design does allow the identification of correlations that have the potential to be selected.

The authors also claim that "Thus, epigenetic variation likely contributes to adaptation through the same mechanism as genetics and not through inheritance of environmentally induced epimutations." (lines 110-111) but don't provide a compelling reason. They claim that the patterns of DMCs could be explained by "selection of standing epigenetic variation instead of the selection of de novo acquired epimutations" (106-107 and 127-129) which seems problematic for two reasons: the most obvious is that regardless of whether the epialleles are already present in the population or induced, the pattern shows a correlation with fitness so DNA methylation is associated with adaptation. The authors seem to be arguing against induced epi-alleles for some undescribed reason. The second problem with this reasoning is that the authors have the data to examine whether the methylation was present in the ancestral population. Its not clear why they have argued that the variation was there already when they just spent several paragraphs describing how the patterns are different in the selected populations. If that variation was there for many of the loci, but perhaps in lower frequency, they should be able to find it for some portion of the loci in the ancestral plants that were sequenced. Otherwise, there is no reason to believe that the methylation patterns behave the same as genetic loci. Overall reduction of DNA methylation does not mean necessarily that what's left is a subset of what was there or preclude novel patterns (lines 201-203).

RESPONSE: We thank Reviewer 1 for bringing this to our attention. We now realize that the arguments for the role of standing epigenetic variation were not as clear as we previously thought (see also comment of

Reviewer 3). We agree that, regardless of the mechanism, DNA methylation is associated with adaptation. We previously argued against acquired *de novo* epimutations because it seemed unlikely that random changes occur several times independently, that is in three replicated selection experiments, at the same place. However, specific environmental conditions can lead to similar epigenetic changes and while the vast majority of them is not heritable (e.g. Wibobo *et al.*, 2016), some may be. Although we consider *de novo* epimutations during the selection experiment as unlikely, some have likely arisen prior to selection during the hybridization to generate the founder population of the RILs. Thus, there may have been a high epimutation rate in the pedigree of our populations, and the regions altering their methylation state may be non-random and biased towards one parent (e.g. Greaves *et al.*, 2012). As we cannot disentangle the origin of epimutations, we changed our interpretation accordingly and discuss these issues in more detail in the revised manuscript.

From the first paragraph: The authors should acknowledge that heritable variation may come in more than two so-called “flavors” considering that they have only just investigated 2.

RESPONSE: We changed the first paragraph according to the journal’s guidelines (Nature Communications requires an abstract instead of an introductory paragraph). As a consequence, we also do not mention the two flavors anymore.

Line 33 This description of the experiment is confusing. The phrase “reveal the outcome of selection” implies that the source of selection is understood (eg drought, heat, competition etc), but the authors don’t describe the selective environment. It could be better to more generally describe “response to novel natural field environments” or something along those lines. Its also not clear here or elsewhere what is meant by “only two genotypes (CVL39 and CVL125) dominated the populations”. Were these two equally represented across all of the populations? Or did one dominate some and the second dominate the others? This type of disparity would indicate that perhaps the two were differentiated by habitat type (biotic or abiotic). We haven’t time to go back to the original study, so this important information should be provided.

RESPONSE: We included a more detailed description of the original study and its results. On average, CVL39 and CVL125 were similarly present (CVL39: 47% and CVL125: 43%) in the selected populations, which we tested at that time (which were the same that we used in the present study). However, percentages within each landscape varied (CVL39: 33%, 87%, and 21% and CVL125: 60%, 10%, and 59%). Together, these two genotypes clearly dominated in each of the selected populations (93%, 97%, and 79%).

Line 47 resequencing individuals from 1 of the genotypes will not tell you necessarily that the same thing hapend in all individuals of that genotype or if the findings would be the same for the other genotype. e.g. the single random mutation in MEE57 in one of the mutation accumulation lines changed genome-wide methylation of otherwise Col wild type. Incidentally, didn’t you also sequence 24 individuals (line 66) that would shed light on this question?

RESPONSE: Unfortunately, we could not resequence more individuals at that time. Clearly, we cannot predict the genotype of the individuals that were not sequenced. However, we are confident that mutations strongly affecting the DNA methylation machinery would have been detected while analyzing the data. Additionally, in order to detect significant differences in DNA methylation, the effect of a potential mutation must be consistent in all three selected populations. Thus, it would have had to occur three times independently. However, this is very unlikely even if mutation rates would have been elevated during selection for some unknown reason.

lines 54-61 These arguments about phenotypes and fitness are quite convincing and a real strength of the manuscript.

RESPONSE: Thank you for this positive comment.

line 89 Its unclear what this sentence means: “Only 20/23 terms were significantly enriched in at least 50 (out of 121) parameter combinations in CVL39 and CVL125. . .”

RESPONSE: We clarified this sentence.

line 131-141 This description and notation are very confusing as written.

RESPONSE: We tried to improve the description and simplified the notation.

line 137 which plants are defective in DNA methylation pathways?

RESPONSE: We extended this sentence in the revised manuscript. It's hopefully clear now that these were results from another study.

line 143 there's no reason to expect a global pattern of DNA methylation association with expression. We know that a lot of methylation is stochastic and seemingly non-functional. The point is to find the few that might be functional.

RESPONSE: We included references which show that DNA methylation and gene expression are globally not associated.

lines 145-165 This analysis of methylation and expression of At2g06002 is definitely of interest and more of this would be welcome. However, given that the methylation was measured on different plants than the expression, its not clear how accurate these findings might be?

RESPONSE: Concerning *At2g06002*, the findings are likely accurate because we can see the same correlation across many different accessions. To substantiate our findings, we performed and included an additional experiment, in which we monitored expression of *At2g06002* (and a few other genes) in all individuals of CVL125 that were used for the original DNA methylation analysis (Supplementary Figure S3 and Supplementary Table S12). With this experiment, we could confirm the correlation between methylation and expression in all individuals tested. For example, the two selected individuals in which the allele was methylated as in the ancestral individuals also had *At2g06002* expression levels similar to the ancestral individuals.

lines 166-175 Again, very interesting discussion, but what are the actual patterns of DMCs? Its difficult to decipher which expression patterns were associated with changes in methylation. The authors only suggest that FT is not associated with DMCs.

RESPONSE: We included a supplemental table that summarizes the associations between DNA methylation and gene expression shown in Figure 3e (Supplementary Table S13).

lines 186-190 The argument made here that " Although we cannot fully exclude the possibility that segregating alleles partly contributed to the observed phenotypic changes or differences in methylation patterns, it is very unlikely that they underlied these changes because none of the affected genes (or genes near an intergenic SNP) have a known role in processes related to flowering time, plant stature, or DNA methylation. " is not so convincing considered how little we actually know about how the genome functions and translates into phenotype.

RESPONSE: We interpreted these findings more cautiously and suggest that it is unlikely but might nonetheless be possible that these SNPs contributed to the phenotype.

Reviewer 2

The manuscript from Schmid and colleagues tackles an interesting question: Whether epigenetic variation can be acted upon by natural selection and thereby contribute to adaptation. The authors make good use of existing populations of recombinant inbred lines that had been previously subjected to selection in a simulated fragmented habitat. The goal of the experimental design was to make good seed dispersal essential and reduce competition between genotypes. The authors examined the ancestral population and three selected, descendant populations. The ancestral population contained nineteen genotypes, and the descendant populations had been subjected to five generations of selection. The prior study found that two genotypes dominated the descendant populations. This is summarised in the submitted manuscript (and clearly, appropriately attributed), but is slightly difficult to follow. The authors might consider moving Fig 1A to a separate figure and expanding it - it took me some time to understand that their new analyses focused on individuals of the two dominant genotypes but that these were not the only genotypes in the populations.

RESPONSE: We thank Reviewer 2 for the encouraging comment. We had included such an extended version of Figure 1A (Extended Data Figure 1) and referenced it in the main text and the legend of Figure 1a.

The authors present reasonably persuasive evidence that epigenetic diversity is reduced in the genotypes that dominate the descendent populations relative to individuals of those genotypes, correlating with phenotypic differences (Fig 1B, C). This result is key to all subsequent data interpretation. I would consequently like the authors to comment on the difference in the number of ancestral and dependent individuals used to calculate the mean pairwise distance between DNA methylomes. Only four ancestral individuals were used, compared with 8 per genotype (16 total) descendants. I am unfamiliar with the mean pairwise distance statistic; how does it take account for differing numbers of individuals and what possibility is there that the relative diversities are artefacts of this?

RESPONSE: The Mean Pairwise Distance (MPD) does not take account for differing numbers of individuals in a specific way. However, MPDs are *per se* uncorrelated to the number of individuals (see for example Allan *et al.* 2013, Ecology 94: 465-477 or Vellend *et al.* 2010, in Biological Diversity: Frontiers in Measurement and Assessment). We tried to clarify this in the methods section of the revised manuscript.

The authors demonstrate that their results indicate selection has acted upon epigenomic variation that existed in the ancestral population, rather than on environmentally induced epigenomic variation. They go on to focus upon a candidate differentially methylated region around At2g06002 that may impact flowering time, using phenotypic and epigenomic data from 54 Arabidopsis accessions to further assess the region's significance. This approach is convincing. Can the authors please explain, though, whether the two variant DNA methylation states found in the descendent populations were detected in any ancestral individuals?

RESPONSE: We showed the DNA methylation state of all individuals in Figure 3b. In CVL125, all individuals of the ancestral population of CVL125 were methylated and most (but not all) individuals of the selected populations were demethylated. In CVL39, all individuals were demethylated (irrespective of “ancestral vs selected”). It is important to note that our ancestral populations also grew for 2-3 generations in the common/controlled environment. Thus, their DNA methylation pattern is not completely identical to the one of the founder population of the original experiment. As detailed below in a response to Reviewer 3, the methylation state of *At2g06002* in CVL125 was originally (at the initial hybridization) likely methylated. Whether all individuals of the founder population were still carrying a methylated allele or not cannot be answered with our data. We include a more detailed discussion on the methylation state of this gene in the revised version of the manuscript (please see below our answer to the comments of Reviewer 3).

Also, how do they reconcile the fact that in one dominant descendent genotype expression of At2g06002 was increased and in the other dominant descendent genotype it was increased, but both descendent genotypes have delayed flowering? I think their argument needs some rephrasing in that section.

RESPONSE: We thank Reviewer 2 for pointing out this inconsistency. We noticed that the comparison of CVL39 with selected CVL125 populations was confusing in this section because both had low methylation levels. We now compare them to the ancestral population of CVL125 which has a high methylation level.

Reviewer 3

This paper argues that selection on epigenetic variation has contributed to phenotypic change in two recombinant-inbred line populations. I first heard these results presented in a meetings years ago, and it would be great if they were finally published. Presumably the authors have suffered from the tendency in this field of declaring results as either trivial and well-known or potentially very interesting but unproven. My personal opinion is that these result are both interesting and novel, that the level of proof is good enough, but that we still don't know what is going on. I have a few concerns, though. . .

First, I would moderate claims about the importance of selection on epigenetic variation. You have not really demonstrated (as stated in the Abstract) that “epigenetic plays a role in fast adaptive responses”, but rather that it CAN play a role. Similarly, on line 25, you mention “a considerable contribution of induced epialleles”. I'm sure there are other such statements, and I don't see how they can be supported without

explicitly comparing the epigenetic to the genetic contribution. You are actually in a position to do so: in your selection experiment, the several traits changed dramatically because of selection among RILs (only two survived). Then you also show that the same traits changed within RILs. Which contributed most to the change in phenotype?

RESPONSE: We thank Magnus Nordborg for his insightful comments and agree that some of our statements should be worded with more caution. We moderated our claims about the importance of selection of epigenetic variation. As explained in an answer above, differences in flowering time between CVL39 and CVL125 were highly significant as well (Supplemental Table S1). Indeed, the effect of genetic variation was clearly larger than the effect of epigenetic variation. For example, differences between the two RILs explain around 15% of all variation in the flowering time proxies (day of bolting and number of rosette leaves at bolting), whereas the contrast for selection only explains around 7%. Likewise, demethylation of *At2g06002* seems to delay flowering in CVL125. In CVL39, *At2g06002* was demethylated in all populations, but CVL39 still flowered earlier than CVL125. We discuss this in more detail in the revised version of the manuscript (also with the results from the original experiment).

Second, I'm confused by your arguments about standing epigenetic variation. Surely the original experiments were bulked from single-seed descent RILs? Does your explanation work, quantitatively?

RESPONSE: To our knowledge, seeds for the original experiment were multiplied from a pool of seeds obtained from the stock center. Thanks to the comments of Reviewers 1 and 3, we realized that our conclusions regarding standing epigenetic variation were less clear than we previously thought (see also our response to Reviewer 1). We had argued against *de novo* acquired epimutations because it seemed unlikely to us that random changes occur several times independently at the same place. However, considering that the RILs were generated using pure breeding lines, all variation observed in any of the populations must have been acquired at some stage. Thus, even if *de novo* mutations during the selection experiment were unlikely, they might have caused the variation in the founder population, e.g through hybridization. We cannot disentangle the contribution of *de novo* epimutations that occurred randomly, were caused by hybridization, or were environmentally induced. We changed our interpretations accordingly and discuss the potential origins of epigenetic variation in more detail in the revised version of the manuscript.

Third, what would account for the big difference between the two RILs? It may be worth considering our recent results (Dubin et al. 2015, Kawakatsu et al. 2016) demonstrating the existence of major trans-acting variation affecting methylation. Cvi, in particular, is known to have very different methylation levels, and the alleles responsible for this must be segregating in the Cvi-Ler RIL population. It is thus possible that epimutation rates differ between lines, and also that the entire genome is not at steady state w.r.t. the genetic background.

RESPONSE: Thank you for pointing out your recent results. They were indeed very helpful to discuss the sources of epigenetic variation in our experiment. From all major trans-acting loci (*AGO1*, *AGO9*, *NRPD1B*, *CMT2*, *MBD3*, and *MET1*), all come from the *Ler* background in both RILs, except for *NRPD1B*, which stems from *Ler* in CVL39 but from *Cvi* in CVL125. Interestingly, the *Cvi* variant seems to be quite distinct from the *Col-0* and the *Ler-1* variants (1001genomes.org/polymorph). Given that *NRPD1B* is required for RdDM at non-CG sites, we now discuss the possibility that the more pronounced loss of methylation at CHG sites in CVL125 compared to CVL39 could be partly due to this locus (Fig. 2e). Likewise, the loss of DNA methylation at *At2g06002* might be related to reduced RdDM activity in CVL125 which is discussed in the revised version of the manuscript.

Fourth, I'm very confused by what you mean by a DMC. Your premise is that methylation states are inherited. Well, then the methylation states should be either 0 or 1 (or possibly 1/2). Like a SNP. In each replicate. Looking at your supplementary tables 3 and 4, you compute averages of methylation percentages, some of which are certainly not 0 or 1. This is not internally consistent. Especially, how can you argue that CHG and CHH sites are part of epigenetic selection, when the methylation fractions per site are far from 0 or 1...? I should emphasize that I don't know what is going on either, but it does seem that your data rule out methylation generally acting as a "fifth base", which is kind of what you are arguing it does... This is must be addressed somehow, if only by acknowledging that there is a problem... More constructively, can't you explicitly test for methylation variants that appear to behave like SNPs?

RESPONSE: Indeed, many of the differences are not 0/1. In Supplementary Tables 3 and 4, the averages can simply be different from 0 or 100 % because it was averaged over all individuals. However, the values still imply that even methylation levels of individuals were not 0/0.5/1 (which is true). It is also not entirely clear to us what causes this. While to a certain extent it may be technical noise, we believe that it is majorly due to differences in DNA methylation levels of different tissues and cell types (we sampled inflorescences, which contains male and female gametophytes and many non-reproductive cell types). If “fifth base”/“like SNP” means that the DNA methylation must be stable across all tissue types of the individual, we can test for “SNP-like behavior” by extracting all cytosines, which are almost 0 or almost 100 percent methylated across all individuals tested (excluding the cytosines that are entirely demethylated in all individuals). With this approach, there are almost no cytosines that behave like a fifth base/SNP (0.0013 % and 0.0017 % in CVL39 and CVL125, respectively, therefrom almost all are in the CG context). However, this approach may be too strict because many tissue types do not contribute to the next generation. To test for SNP-like cytosine methylation, it may be better to compare different tissues/cell types which contribute to the next generation to each other (within individuals). In this sense, one could imagine that loss of DNA methylation at such a “SNP-like” cytosine only happens in specific but not all cell types. For example, rates of a change in DNA methylation may be lower in cells of the shoot apical meristem than in other cells. Thus, the average DNA methylation in a mixed tissue may be lower than 1, but as long as cells in the meristem keep their DNA methylation state, the methylated variant could be kept in the reproductive lineage as well. Interestingly, *MET1*, *DRM2*, and *CMT3* are expressed at (very) high levels in the stem cells from the shoot apical meristem (97th, 90st, and 78th percentile, data from <https://doi.org/10.1101/gad.289397.116>). We discuss these findings now as well in the revised manuscript.

Reviewer 4

This manuscript by Schmid et al provides evidence that epigenetic variation can be subject to natural selection and lead to adaptive phenotypic changes in a changing environment. The study is based on a previously published selection experiment by the same authors (Fakheran et al PNAS 2010), where starting from recombinant inbred lines (RILs) natural selection in a simulated disturbed habitat over five generations was applied, and phenotypic differences were recovered (growth, flowering). The present study aims to identify the underlying molecular variation that is responsible for these phenotypic differences. Since the starting material (the RILs) was supposedly genetically identical, the authors focused on variation at the epigenetic/DNA methylation level. By analyzing two independently selected lines derived from two different RILs, the authors identify through methylome and transcriptome analysis a large number of differential DNA methylation regions (DMR) that may account for the phenotypic differences. In one of the lines they identify a DMR that is associated with differential expression of a non-coding RNA and at the population level is associated with altered flowering time at 10°C. However, this was not affected in the other line, where instead they find changed expression and methylation in a number of circadian clock and flowering time genes. This suggests that the molecular basis of the phenotypic changes may not be the same in the two lines. The authors conclude that the DMRs were selected from standing variation in the parental RILs (residual heterozygosity for DMRs).

The question of whether epigenetic variation can be selected for in natural variation is highly timely and of high interest in the research community. The study is conducted elegantly and thoroughly and the manuscript reads well. However, the underlying hypothesis is that there is no genetic variation left in the RILs and thus any phenotypic changes must be caused by epigenetic changes, - the authors themselves provide evidence that this is not the case. In a genome resequencing effort that covers only one of the two selection lines under study, they identify 14 SNPs, most of which were present as standing variation also in the parental RIL line. This finding is brushed away with the statement that there is no known function for the affected genes in the relevant processes. How can the authors know this? It remains possible that this genetic variation at least contributes to the phenotypically selected traits. The authors should provide additional evidence to strengthen their case (genetic vs. epigenetic basis) or phrase their claims more carefully. For example, l. 17 of the abstract is not correct and needs to be rephrased.

RESPONSE: Although it unlikely plays a role, Reviewer 4 is correct that we cannot completely rule out that these few SNPs contributed to phenotypic traits or even the epigenetic diversity itself. We corrected the abstract and moderated the claims about the absence/insignificance of genetic variation.

The design of the -omes-experiments is not symmetrical. While for the methylomes extensive data from 3 replicate experiments of both selected lines are collected (CVL39, 125; D1, D5, D6), the transcriptome analysis is performed only for D1 of CVL39 and 125. The genome resequencing is only done for CVL39; D1, D5, D6). As the most important criterion in the end appears to be differential gene expression, wouldn't it make more sense to analyze also D5 and D6 and then look for common differentially expressed genes? Interpretation of the data should also bear in mind that the transcriptome data give only a snapshot of one particular developmental stage/time of day.

RESPONSE: We agree that it would be best to have transcriptome data for all populations, but the material we still have available is not suited for transcriptome studies (the design was originally not symmetrical because covering enough individuals with a fully symmetrical design was not possible at the time of the experiment). We discuss this and the “snapshot” nature of the transcriptome (and DNA methylation) data in greater detail in the revised manuscript. We also performed and included an additional experiment, in which we monitored expression of *At2g06002* (and a few other genes) in all individuals of CVL125 that were used for the DNA methylation analysis (Supplementary Figure S3 and Supplementary Table S12). With this experiment, we could confirm the correlation between methylation and expression in all individuals. For example, the two selected individuals in which the allele was methylated as in the ancestral individuals indeed had *At2g06002* expression levels similar to the ancestral individuals.

Minor comments: l. 148: “was higher than...” - how much higher?

RESPONSE: Included as requested. The log fold change was 1.2 (2.3 times higher on linear scale).

l. 158: flowering time at 10°C appears to be a rather exotic condition. Surely, Ref. 24 or other references must provide flowering time results under various conditions (including more typical ones). Is this the only one where an association was found or was this one selected for a specific reason? This should be stated/explained more clearly.

RESPONSE: The original data from Atwell *et al.* 2010 include a wide range of measurements and conditions. We included a table with all data underlying these analyses (Supplemental Table S15). For flowering time there are data for 10°C, 16°C, and 22°C. However, there was only a significant difference at 10°C. We now refer to these data in the revised manuscript.

l. 163: Is the expression of FIP1 affected?

RESPONSE: In the microarray data, there are no significant differences. However, in the additional experiment we performed, we also tested the expression of *FIP1* in all CVL125 individuals that were used for the DNA methylation analysis. Based on these results, *FIP1* is clearly expressed at a higher level in the selected individuals ($P = 0.0019$, fold-change = 1.7, see also Supplementary Figure S3 and Supplementary Table S12). We included this new data in the main text of the revised manuscript.

Fig. 1B: Do these data stem from the 2nd or 3rd generation?

RESPONSE: The data are from both generations. To show both generations in one figure, we used the differences to the “within generation averages”. We now state this more clearly in the figure legend.

Fig. 2A: It took me a while to understand the Figure. Maybe the relevant part of the Figure could be highlighted to make it more clear?

RESPONSE: We thank Reviewer 4 for the suggestion and highlighted the part which illustrates clustering.

Reviewers' Comments:

Reviewer #1:

Remarks to the Author:

Schmid et al have done a very nice job of thoughtfully revising the manuscript in response to the reviewers' comments. It is a nice contribution to some emerging questions on the importance of epigenetics in several different aspects of ecology and evolution. The information about correlation between methylation and expression, especially the methylation machinery (RdDM etc) and flowering time genes will be extremely useful. I was also glad to see some inclusion and discussion of the ecological epigenetics literature in this version (eg refs 5, 6, 15, 16). A few points of clarification may help to further enhance the reach of the paper to this audience in particular:

Clarification of the concept of induced epialleles on a couple of fronts is important. Eg. Cortijo et al (2014) does a beautiful job of showing epiallelic contribution to phenotypic variance, but these epialleles were not "induced" (Line 21), they were the product of the DDM1 mutation and then bred to be homozygous across epi-loci. It's a small distinction, but it matters when considering the source of epiallelic variation. This can be fixed by deleting the word "induced".

Further, I do not agree with the statement "environmentally induced epimutations, which are often considered directional" which you have attributed to our review in Ecology Letters (ref 16). Directional in this sense implies that there is a so-called "appropriate response" induced to whatever stimulus is present. Instead, we argue that the environment can induce epigenetic changes which MAY be useful for adaptation by providing phenotypic variance upon which selection can act (see also Rapp & Wendel 2005 New Phytologist; Richards, Verhoeven & Bossdorf 2012 in Wendel's Plant Genome Diversity). This is an important distinction because many people are skeptical of the importance of epigenetics in evolution and quickly dismiss the ideas because they rely on the concept of "directed mutations," so the second part of this statement ("which are often considered directional") should be removed.

Along these lines, the authors have mentioned the reprogramming through hybridization (eg. line 293), which could be explored more (see again Rapp and Wendel 2005, Salmon et al 2005 Mol Ecol; the 'genomic shock' concept of Barbara McClintock) in the context of these results. Further, the so-called reprogramming through gametogenesis (275-276) is overstated since it has been argued to be incomplete. E.g discussed in Robertson and Richards 2015 NGI:

Feng S., Jacobsen S.E., Reik W., Epigenetic reprogramming in plant and animal development, Science, 2010, 330, 622-627;

Wei Y., Schatten H., Sun Q.-Y., Environmental epigenetic inheritance through gametes and implications for human reproduction, Human reproduction update, 2014, dmu061.

Another point of confusion that should be clarified: In Lines 62-69, the authors seem to equate within RIL variation as epigenetic and between RIL variation as genetic, when both sources of variation (to different degrees) are possibly occurring in each. This is even more confusing since first the authors say that the epigenetic differences were half the genetic differences, but then "that this relatively large contribution of epigenetic relative to genetic variation reflects the fact that the genetically caused phenotypic variation of populations growing in the dynamic landscapes was strongly reduced due to selection (lines 67-70)". I would suggest removing any reference to genetic versus epigenetic here and just report within line versus across line phenotypic variance. This works better to transition to the next paragraph which starts with "To investigate whether these heritable phenotypic changes were paralleled by changes at the level of DNA methylation..."

Finally, it is interesting that the authors argue that epigenetic variation may be more important in genetically diverse populations since the opposite argument has been made in of some the recent literature, and in more detail. This alternative perspective of the opposing arguments should be

mention (see arguments in Richards et al 2017 Ecol Lett: Rendina Gonzalez et al. 2016; Spens & Douhovnikoff 2016; Verhoeven, K.J.F. & Preite, V. 2014. Epigenetic variation in asexually reproducing organisms. *Evol.*, 68, 644–655).

Reviewer #2:

Remarks to the Author:

The authors have addressed all of my comments appropriately and I am in favour of publication.

Reviewer #3:

Remarks to the Author:

Sorry about the delay. I am happy with the way you have addressed my comments (and those of the other reviewers). There are still some statements that are stronger than I would have made them, but you are the authors, not I. The paper is much better, imho. At this point I really hope this can be published so everyone can join in the argument!

One more thing: you are missing an opportunity to cite Johannssen (1903)

https://en.wikipedia.org/wiki/Wilhelm_Johannsen#cite_ref-3

It's a classic and absolutely relevant here.

Reviewer #4:

Remarks to the Author:

The authors have made a very thorough effort to address my comments and concerns. The manuscript is now suitable for publication.

Detailed Response to the Reviewers' Comments

We would like to thank all reviewers for their interest in our work and the efforts they put into providing constructive criticisms and insightful comments. Below we provide a point-by-point response to all comments. To facilitate reading, the reviewers' comments are in italics, while our responses are in roman and indicated by **RESPONSE**.

Reviewer 1

Schmid et al have done a very nice job of thoughtfully revising the manuscript in response to the reviewers' comments. It is a nice contribution to some emerging questions on the importance of epigenetics in several different aspects of ecology and evolution. The information about correlation between methylation and expression, especially the methylation machinery (RdDM etc) and flowering time genes will be extremely useful. I was also glad to see some inclusion and discussion of the ecological epigenetics literature in this version (eg refs 5, 6, 15, 16). A few points of clarification may help to further enhance the reach of the paper to this audience in particular:

RESPONSE: Thank you for the positive feedback.

Clarification of the concept of induced epialleles on a couple of fronts is important. Eg. Cortijo et al (2014) does a beautiful job of showing epiallelic contribution to phenotypic variance, but these epialleles were not "induced" (Line 21), they were the product of the DDM1 mutation and then bred to be homozygous across epi-loci. It's a small distinction, but it matters when considering the source of epiallelic variation. This can be fixed by deleting the word "induced".

RESPONSE: In a genetic sense, we think that it is appropriate to call these changes "induced" because they did not arise spontaneously but were caused by a mutation. In the context of epiRILs this can be regarded as a "treatment" that greatly increases epigenetic variation. To clarify this, we now write "genetically induced" instead of "induced".

Further, I do not agree with the statement "environmentally induced epimutations, which are often considered directional" which you have attributed to our review in Ecology Letters (ref 16). Directional in this sense implies that there is a so-called "appropriate response" induced to whatever stimulus is present. Instead, we argue that the environment can induce epigenetic changes which MAY be useful for adaptation by providing phenotypic variance upon which selection can act (see also Rapp & Wendel 2005 New Phytologist; Richards, Verhoeven & Bosdorf 2012 in Wendel's Plant Genome Diversity). This is an important distinction because many people are skeptical of the importance of epigenetics in evolution and quickly dismiss the ideas because they rely on the concept of "directed mutations," so the second part of this statement ("which are often considered directional") should be removed.

RESPONSE: We would like to apologize for the misattribution and thank you for the clarifications. To avoid any confusions, we removed the second part of the statement and refer the reader to Richards *et al.* 2017 for further details.

We agree that it is very important to distinguish between "Epimutation" and "Environmental Induction" in a way you and your colleagues did in Richards *et al.* 2017 (summarized in Figure 1) and that "Environmental Induction" should not be confused with the concept of "directed mutations". However, we find it potentially misleading to use "Environmental Induction" to refer to changes in epimutation rates under particular environmental conditions because "induction" seems to imply some sort of directionality. Thereby, "Environmental Induction" would result in a at least partly reproducible set of changes in DNA methylation, which could be understood as directional. Whether these changes are then useful for adaptation or not is a different question.

Along these lines, the authors have mentioned the reprogramming through hybridization (eg. line 293), which could be explored more (see again Rapp and Wendel 2005, Salmon et al 2005 Mol Ecol; the 'genomic shock' concept of Barbara McClintock) in the context of these results.

RESPONSE: Thank you for the suggestion. Indeed, the discussion of whether and how hybridization may partly reprogram the epigenome is highly interesting. We previously only cited studies which are based on genome-wide data in *Arabidopsis thaliana* because we think that these are the most relevant for our study system (interesting in this context is also Göbel *et al.* 2018, “Robustness of Transposable Element Regulation but No Genomic Shock Observed in Interspecific Arabidopsis Hybrids”, *Genome Biology and Evolution*). However, given the constraints in overall text size and number of references, we are concerned that we cannot discuss this topic in sufficient detail.

Further, the so-called reprogramming through gametogenesis (275-276) is overstated since it has been argued to be incomplete. E.g. discussed in Robertson and Richards 2015 NGI, Feng S., Jacobsen S.E., Reik W., Epigenetic reprogramming in plant and animal development, Science, 2010, 330, 622-627, Wei Y., Schatten H., Sun Q.-Y., Environmental epigenetic inheritance through gametes and implications for human reproduction, Human reproduction update, 2014, dmu061.

RESPONSE: We respectfully disagree that this is overstated. Even if the resetting is incomplete, it certainly limits inheritance of environmentally induced epigenetic variation. We think that we worded the statement carefully enough by writing that “... inheritance of environmentally induced epigenetic variation MAY be limited and restricted to certain regions of the genome”.

Thank you for pointing out these reviews. However, we would like to emphasize that all eight studies cited to support our statement were not referred to in any of these reviews, either because the studies were newer than the review (Feng *et al.* 2010) or they do not discuss epigenetic resetting in plants (Wei *et al.* 2014) in great detail (Robertson and Richards 2015). To the best of our knowledge, recent research clearly indicates that epigenetic reprogramming in plants is more prominent than it was thought to be a few years ago. Given the limited space and number of references we can include, we decided to limit citations on this topic to more recent studies and reviews.

Another point of confusion that should be clarified: In Lines 62-69, the authors seem to equate within RIL variation as epigenetic and between RIL variation as genetic, when both sources of variation (to different degrees) are possibly occurring in each. This is even more confusing since first the authors say that the epigenetic differences were half the genetic differences, but then “that this relatively large contribution of epigenetic relative to genetic variation reflects the fact that the genetically caused phenotypic variation of populations growing in the dynamic landscapes was strongly reduced due to selection (lines 67-70)”. I would suggest removing any reference to genetic versus epigenetic here and just report within line versus across line phenotypic variance. This works better to transition to the next paragraph which starts with “To investigate whether these heritable phenotypic changes were paralleled by changes at the level of DNA methylation...”

RESPONSE: Indeed, we implicitly equated within RIL variation as epigenetic and between RIL variation as genetic. As we show that there are very few genetic differences within at least one of the RILs (resequencing of CVL39 individuals), we think that it is justified to assume that the within RIL variation is mostly of epigenetic nature. However, we agree that there could be additional epigenetic variation - which is not caused by genetic variation - between RILs on top of the genetic variation. To clarify this, we extended “The variance of traits related to reproduction and fecundity explained by epigenetic differences within each RIL was almost half the variance explained by genetic differences between the two RILs” by adding “(assuming that the majority of the phenotypic differences between RILs were due to genetic differences and not due to epigenetic differences)”.

This equation and comparison was suggested by reviewer 3 and we agree that this is an interesting comparison because, even though there is possibly residual genetic variation within RILs and epigenetic variation between RILs, we cannot think of a different experimental setup that would allow for such a comparison in a more accurate way.

Finally, it is interesting that the authors argue that epigenetic variation may be more important in genetically diverse populations since the opposite argument has been made in of some the recent literature, and in more detail. This alternative perspective of the opposing arguments should be mention (see arguments in Richards et al 2017 Ecol Lett: Rendina Gonzalez et al. 2016; Spens & Douhovnikoff 2016; Verhoeven, K.J.F. & Preite, V. 2014. Epigenetic variation in asexually reproducing organisms. Evol., 68, 644–655).

RESPONSE: Thank you for pointing out these alternative perspectives. We clarified the statement, explained it in more detail, and cited alternative perspectives. However, due to constraints in the number of references, we only included Richards *et al.* 2017 and Verhoeven and Preite 2013. Specifically, we changed “efficiently” into “frequently” to be more accurate and added:

“This may be unexpected because it is frequently argued that epigenetic variation may evolutionarily be more important in populations with low genetic diversity and asexually reproducing species (e.g., Verhoeven and Preite 2013, Richards *et al.* 2017). However, although epigenetic variation may be more frequent in genetically diverse species, genetic diversity is much higher as well. Hence, the relative importance of epigenetic variation may still be higher in populations with low genetic diversity and asexually reproducing species.”

Reviewer 2

The authors have addressed all of my comments appropriately and I am in favour of publication.

RESPONSE: Thank you.

Reviewer 3

Sorry about the delay. I am happy with the way you have addressed my comments (and those of the other reviewers). There are still some statements that are stronger than I would have made them, but you are the authors, not I. The paper is much better, imho. At this point I really hope this can be published so everyone can join in the argument!

One more thing: you are missing an opportunity to cite Johannssen (1903)

en.wikipedia.org/wiki/Wilhelm_Johannsen#cite_ref-3

It's a classic and absolutely relevant here.

RESPONSE: Thank you. We included a reference to Johannssen 1903.

Reviewer 4

The authors have made a very thorough effort to address my comments and concerns. The manuscript is now suitable for publication.

RESPONSE: Thank you.